# Deciphering in-situ surface reconstruction in two-dimensional CdPS$_3$ nanosheets for efficient biomass hydrogenation

Marshet Getaye Sendeku[1,2,3,8], Karim Harrath[4,8], Fekadu Tsegaye Dajan[3], Binglan Wu[3,5], Sabir Hussain [6], Ning Gao[3], Xueying Zhan[3], Ying Yang [5], Zhenxing Wang [3], Chen Chen[7], Weiqiang Liu[2], Fengmei Wang [1,3] ✉, Haohong Duan [7] ✉ & Xiaoming Sun [1,2] ✉

Steering on the intrinsic active site of an electrode material is essential for efficient electrochemical biomass upgrading to valuable chemicals with high selectivity. Herein, we show that an in-situ surface reconstruction of a two-dimensional layered CdPS$_3$ nanosheet electrocatalyst, triggered by electrolyte, facilitates efficient 5-hydroxymethylfurfural (HMF) hydrogenation to 2,5-bis(hydroxymethyl)furan (BHMF) under ambient condition. The in-situ Raman spectroscopy and comprehensive post-mortem catalyst characterizations evidence the construction of a surface-bounded CdS layer on CdPS$_3$ to form CdPS$_3$/CdS heterostructure. This electrocatalyst demonstrates promising catalytic activity, achieving a Faradaic efficiency for BHMF reaching $91.3 \pm 2.3$ % and a yield of $4.96 \pm 0.16$ mg/h at $-0.7$ V versus reversible hydrogen electrode. Density functional theory calculations reveal that the in-situ generated CdPS$_3$/CdS interface plays a pivotal role in optimizing the adsorption of HMF* and H* intermediate, thus facilitating the HMF hydrogenation process. Furthermore, the reconstructed CdPS$_3$/CdS heterostructure cathode, when coupled with MnCo$_2$O$_{4.5}$ anode, enables simultaneous BHMF and formate synthesis from HMF and glycerol substrates with high efficiency.

Chemical manufacturing industries account for nearly 18 % of greenhouse gas (GHG) emissions[1], about 8% of global energy consumption[2], and the use of fossil fuel contributes to around 85% of the GHG emitted from these industries. In view of this, the valorization of biomass-based substrates to produce high-valued chemicals is considered a game-changer strategy to concurrently reduce the huge reliance on fossil fuel and GHG emission[3]. 5-Hydroxymethylfurfural (HMF), a key intermediate in biomass conversion, can be obtained by dehydration of biomass-derived carbohydrates, such as starch, glucose, sucrose, and fructose[4]. HMF can be upgraded into various high-valued chemicals, such as 2,5-bishydroxymethylfurfural (BHMF) and 2,5-dimethylfuran (DMF). Among these products, BHMF has numerous industrial

[1]State Key Laboratory of Chemical Resource Engineering, College of Chemistry, Beijing University of Chemical Technology, Beijing 100029, PR China. [2]Ocean Hydrogen Energy R&D Center, Research Institute of Tsinghua University in Shenzhen, Shenzhen 518057, PR China. [3]CAS Key Laboratory of Nanosystem and Hierarchical Fabrication, National Center for Nanoscience and Technology, Beijing 100190, PR China. [4]Department of Chemistry, Southern University of Science and Technology, Shenzhen 518055, PR China. [5]Shaanxi Provincial Key Laboratory of Electroanalytical Chemistry, Key Laboratory of Synthetic and Natural Functional Molecule of the Ministry of Education, College of Chemistry & Materials Science, Northwest University, Xi'an 710127, PR China. [6]Tyndall National Institute, University College Cork, Dyke Parade, Cork T12 R5CP, Ireland. [7]Department of Chemistry, Tsinghua University, Beijing 100084, PR China. [8]These authors contributed equally: Marshet Getaye Sendeku, Karim Harrath. ✉e-mail: wangfm@buct.edu.cn; hhduan@mail.tsinghua.edu.cn; sunxm@mail.buct.edu.cn

importance in manufacturing polyurethane foams, resins, and artificial fibers[5], and serves as a vital intermediate in the synthesis of drugs and crown ethers[6,7]. However, its current industrial production still relies on the thermocatalytic route (Fig. 1a), which requires a precious metal (Ru, Pt, Pd, Ir) catalyst, harsh operation conditions of high pressure (28–350 bar), and temperature (403–423 K), as well as utilize $H_2$ gas as reductant, imposing both energy and environmental concerns[8].

The synthesis of hydrogenated products of HMF via heterogeneous electrocatalysis affords a multi-fold advantage since water can be directly utilized as a hydrogen donor at a relatively ambient condition (Fig. 1b). Indeed, electrocatalytic HMF hydrogenation (ECH) in nonacidic media involves a multi-step complex reaction which also requires the dissociation of $H_2O$ on the catalyst surface to produce H* intermediate. The fundamental challenge in the electrochemical conversion of HMF to BHMF is the formation of undesired 5,5 bis(hydroxymethyl)hydrofuroin (BHH) dimer upon a single $H^+/e^-$ transfer at lower overpotential and the competing hydrogen evolution reaction (HER) at higher overpotential[9]. To partly unravel this puzzle, electrode materials bearing Ag and Cu with poor activity for HER are utilized, which practically depict a reasonable selectivity (>85%) to BHMF[10–12]. As those well-performing electrodes still comprise precious metals (such as Ag)[13], the design of ideal catalysts based on earth-abundant elements is still required for practical application.

The rational design of an electrocatalyst for electrochemical conversion of reactants/feedstock requires a full understanding of the reaction process and the key parameters for controlling the electrode surface and interface[14–17]. For instance, the dynamic electrode surface of the heterogeneous electrocatalyst, which serves as the real active site, has been widely studied in kinds of electrocatalysts, like metal phosphides, chalcogenides, and oxides[18–23]. The in situ generated active species under different operating conditions could play a pivotal role in manipulating the adsorption energetics of the most important reaction intermediates, which gives rise to

improved catalytic activity[22,24,25]. In recent years, two-dimensional materials have been widely employed to electro-catalyze the conversion of biomass-based feedstock into high-valued chemicals[26,27]. Inspired by these developments and our interests in the growth of two-dimensional (2D) metal phosphorus trisulphide ($MPS_3$) nanosheet arrays on conductive substrates[28–31], we speculate that the rich sulfur and phosphorus environment in $MPS_3$ could afford a greater degree of structural flexibility to readily undergo surface reconstruction for promoting the product selectivity. As a result, it can be anticipated that numerous kinds of active sites can be exposed to facilitate catalytic reactions, which was seldom realized in previous studies.

Here we show that an in-situ surface reconstruction of novel $CdPS_3$ nanosheet electrocatalyst triggered by electrolyte facilitates efficient HMF hydrogenation to BHMF at room temperature. This reconstructed electrocatalyst exhibited superior performance with a high Faradaic efficiency (FE) of 91.3 ± 2.3 % for BHMF synthesis with the yield reaching 4.96 ± 0.16 mg/h at −0.7 V versus reversible hydrogen electrode ($V_{RHE}$). In situ Raman spectroscopy and comprehensive post-mortem catalyst characterizations evidence that the surface of the $CdPS_3$ nanosheet is covered by a sub-10 nm CdS layer, which gives rise to the formation of $CdPS_3$/CdS heterointerface. We offer experimental evidence to identify the initiators that lead the surface transformation during the ECH process, in which the phosphate buffer saline electrolyte (PBS) plays a key role in facilitating the reconstruction process. Density functional theory (DFT) calculation reveals that the in situ generated CdS layer plays a pivotal role in manipulating the adsorption energetics of HMF* and H*, as well as minimizing the energy barriers of vital intermediate, thus facilitating the HMF hydrogenation process. When the electrocatalysts of $CdPS_3$/CdS and $MnCo_2O_{4.5}$ are coupled into a two-electrode cell, the HMF hydrogenation and glycerol oxidation reactions are realized at a low cell voltage of 1.9–2.1 V to yield two high-valued chemicals simultaneously.

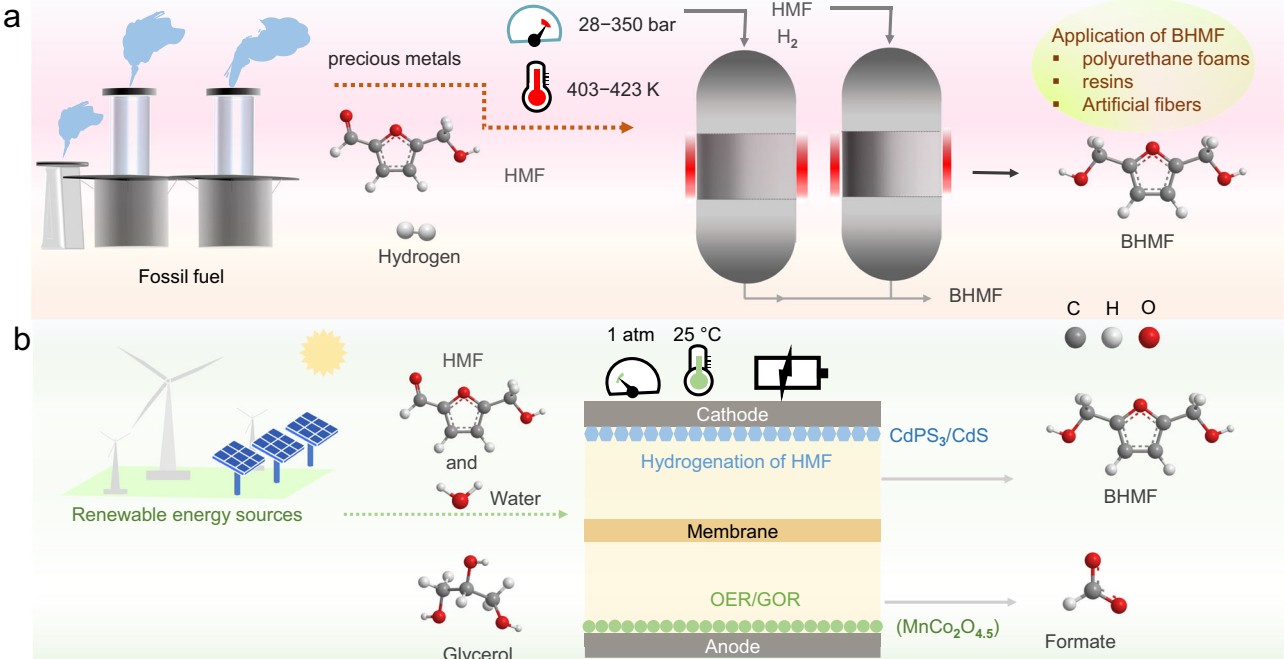

**Fig. 1 | Schematic illustration comparing the electrochemical 2,5-bis(hydroxymethyl)furan (BHMF) synthesis with the conventional thermocatalytic route. a** The conventional thermocatalytic process for BHMF production. The process involves the use of hydrogen supply and precious metal catalysts, and the reaction is carried out under high temperature and pressure. **b** Electrochemical strategy for BHMF synthesis paired with glycerol oxidation under ambient conditions. The 5-hydroxymethylfurfural (HMF) hydrogenation reaction is carried out using water as the source of hydrogen. The extra energy required to derive the chemical transformation can be powered by renewable energy resources.

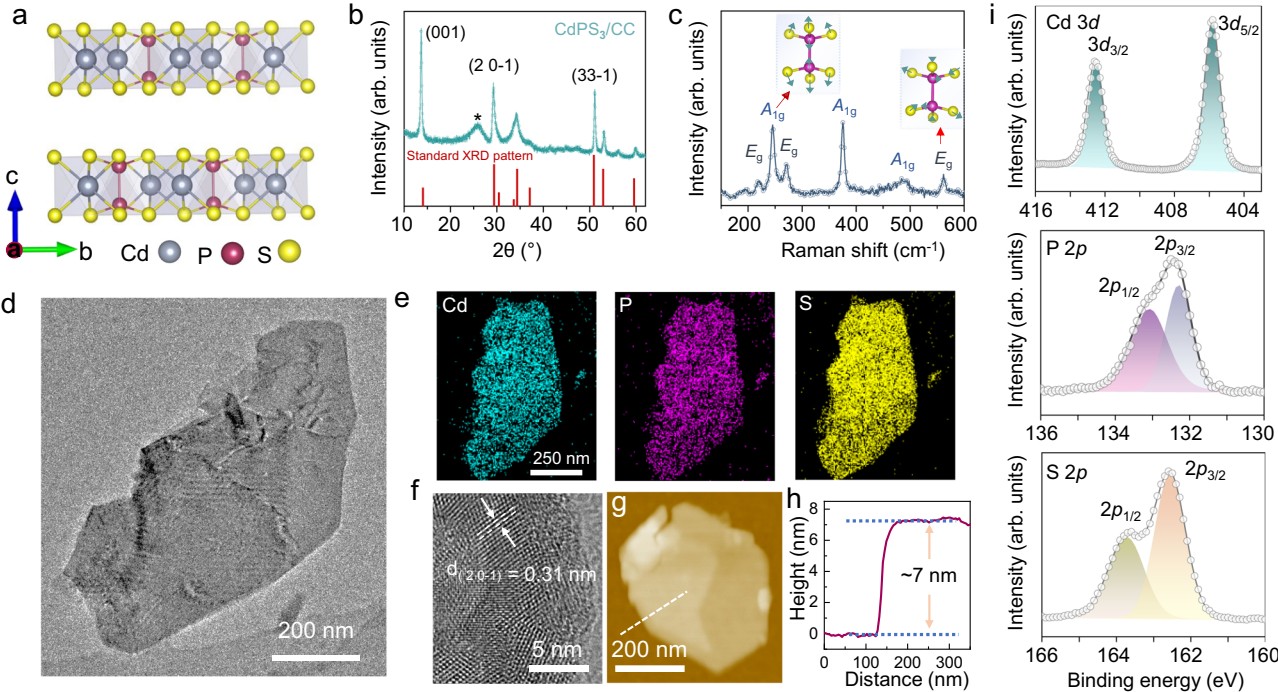

**Fig. 2 | Structural and morphological characterizations. a** Side view of van der Waals layered CdPS$_3$ crystal structure with monoclinic phase (C2/m space group). **b** XRD diffractogram of as prepared CdPS$_3$ on carbon cloth (cyan green) and the standard PDF card (PDF# 33-0243, red). The broad peak at ~27 degree (designated as*) belongs to the carbon cloth substrate. **c** Raman spectrum of CdPS$_3$. Insets, the Raman vibration modes of the P$_2$S$_6$ unit. **d, e** TEM image of one CdPS$_3$ nanosheet (**d**) and the corresponding elemental mapping (**f**). **f** HR-TEM image collected from the CdPS$_3$ nanosheet in (**d**). AFM image (**g**), and the corresponding height profile (**h**) of ultrathin CdPS$_3$ nanosheets. **i** XPS spectra of Cd, P, and S for the as-grown CdPS$_3$ nanosheet.

## Results

### Electrocatalyst synthesis and characterizations

Recently, novel electrocatalysts based on MPS$_3$ have been widely employed for catalyzing several electrochemical reactions, including HER[32], oxygen evolution reaction (OER)[33], carbon dioxide reduction[34], and nitrogen reduction reactions[35], and thus demonstrated their promising performances. With the view that HER is the major competing reaction to electrochemical HMF hydrogenation, we choose CdPS$_3$ as a potential candidate following its poor HER activity demonstrated in pioneer work[36]. CdPS$_3$ crystallizes as a CdCl$_2$-type monoclinic structure (C2/m, $a = 6.218$, $b = 10.763$, $c = 6.867$, $\alpha = 90.0$, $\beta = 107.58$, and $\gamma = 90.0$) with adjacent layers possessing an interlayer spacing of 6.5 Å connected through a weak Van der Waals interaction (Fig. 2a and Supplementary Fig. 2a, b). Here we employ a two-step solvothermal–space confined chemical vapor conversion process to synthesize ultrathin CdPS$_3$ nanosheets (NSs) on a carbon cloth substrate. Briefly, the CdS nanoparticle precursor was first synthesized through a solvothermal method (Supplementary Note I and Supplementary Fig. 2a–c). The Raman spectrum (Supplementary Fig. 2d) show two peaks at ~301 and 600 cm$^{-1}$, which are the longitudinal optical phonon modes in the CdS sample[37]. The as-obtained CdS nanoparticle precursor was then directly converted into CdPS$_3$ NSs in a tube furnace using specially configured silica socket tubes (detailed in Supplementary Fig. 3 and Methods). Note that the CdPS$_3$ nanosheet growth is governed by the temperature and reaction time, in which a temperature and growth time of 420 °C and 20 min, respectively, are required for the completion of ultrathin CdPS$_3$ NS synthesis. The scanning electron microscope (SEM) image and corresponding energy-dispersive X-ray spectroscopy profile show the as-prepared CdPS$_3$ NSs are vertically grown on the carbon cloth uniformly covering the entire surface of the substrate (Supplementary Fig. 4a, b) with the Cd: P: S ratio close to 1:1:3 (Supplementary Fig. 4c). As evidenced from the XRD pattern in Fig. 2b, the diffraction peaks located at 13.51, 29.05,

and 50.86 degrees, which are ascribed to (001), (20-1), and (33-1) planes, respectively, of a monoclinic phase CdPS$_3$ (PDF#33-0243), are clearly observed. It is interesting to note that the intensity ratio of (001)/(33-1) in the XRD pattern of the as-grown sample is much higher (1.67) than the standard value (0.3), implying the preferred growth orientation in the c direction[38,39]. The Raman spectrum was also gathered to reveal the symmetric structure and trace the peaks that emanate from the S$_3$P-PS$_3$ unit of the D$_3$d symmetry group. The Raman peaks assignable to the $A_{1g}$ out-of-plane vibration modes (245.3, 375.7, and 486.8 cm$^{-1}$) and $E_g$ in-plane vibration modes (219.0, 271.0, and 561.6 cm$^{-1}$) are clearly seen (Fig. 2c), which are also consistent to other reports of layered CdPS$_3$ elsewhere[40].

The transmission electron microscopy (TEM) image and elemental mapping clearly depict that Cd, P, and S are uniformly distributed throughout the nanosheets of ~700 nm lateral size (Fig. 2d, e). High-resolution TEM (HR-TEM, Fig. 2f) conducted on the CdPS$_3$ nanosheets display a typical lattice fringe with a spacing of 0.31 nm corresponding to the (20-1) plane, which corroborates well with the XRD result in Fig. 2b. The thickness of ~7 nm (Fig. 2g, h) for single CdPS$_3$ nanosheet is verified by atomic force microscopy (AFM). The surface chemical state of the CdPS$_3$ nanosheet sample was also examined by X-ray photoelectron spectroscopy (XPS, Fig. 2i and Supplementary Fig. 5). The high-resolution XPS spectrum of Cd 3$d$ consists of two peaks situated at nearly 412.6 and 405.8 eV, which corresponds to 3$d_{3/2}$ and 3$d_{5/2}$ of Cd$^{2+}$, respectively. In the XPS spectrum of P and S of the CdPS$_3$ sample, peaks located at 132.3 and 133.1 (P 2$p_{3/2}$ and P 2$p_{1/2}$ of P$^{4+}$), and 162.5 and 163.7 eV (S 2$p_{3/2}$ and 2$p_{1/2}$ of S$^{2-}$) are clearly observed.

### Electrocatalytic hydrogenation of HMF over CdPS$_3$ nanosheets

To evaluate the electrocatalytic activity of CdPS$_3$ NSs towards HMF hydrogenation, we first examined the linear sweep voltammetry (LSV) curves in 0.1 M phosphate buffer solution (PBS, pH~9.2) in the absence

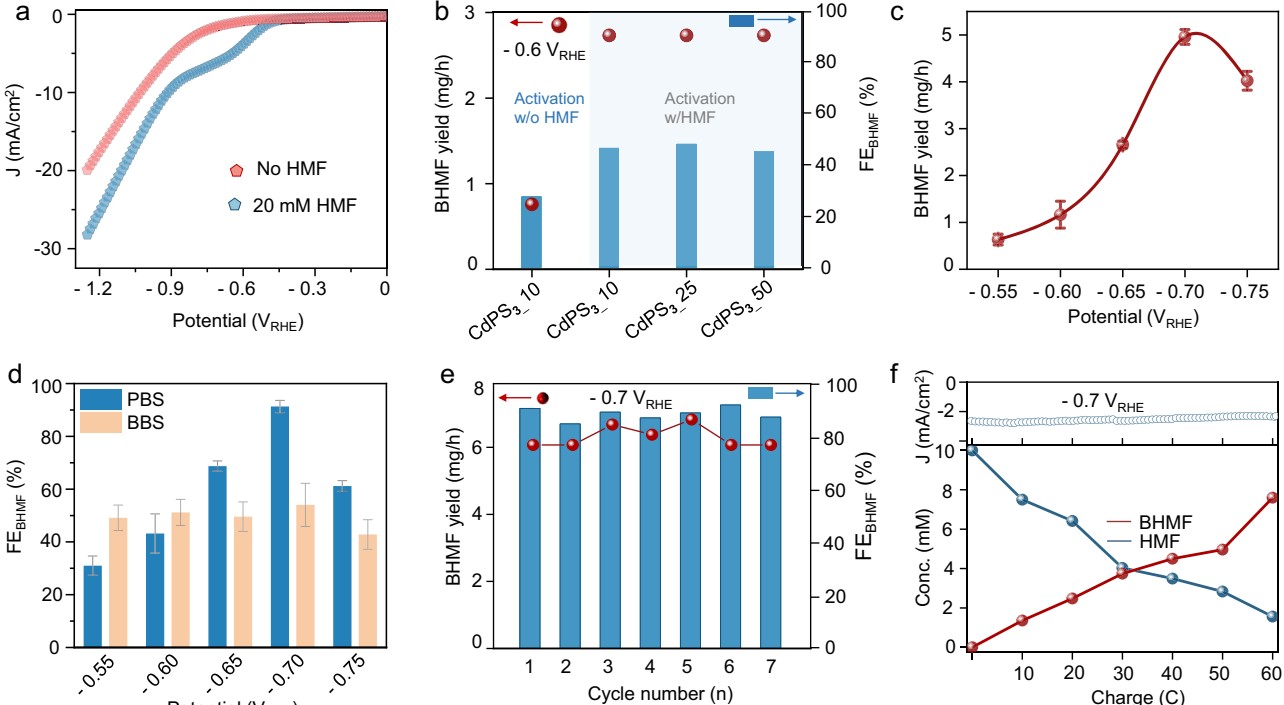

**Fig. 3 | Electrocatalytic activity of CdPS$_3$ catalyst for hydrogenation of HMF to BHMF. a** Polarization curves (without $iR$ correction) of CdPS$_3$ nanosheet electrode in 0.1 M PBS (pH = 9.2) electrolyte with (blue) and without (brown) 20 mM HMF. **b** Comparison of HMF hydrogenation activity of CdPS$_3$ electrodes activated in 0.1 M PBS in the presence and absence of HMF under 10, 25, and 50 CV activation cycles. The chronoamperometric method was employed at −0.6 V$_{RHE}$ to evaluate their performances. **c** The potential dependent BHMF yield of CdPS$_3$ _25 electrode for 10 mM HMF hydrogenation in 0.1 M PBS electrolyte. **d** Potential dependent FE for CdPS$_3$ _25 electrodes in 0.1 M PBS (blue) and 0.1 M BBS (orange). Error bars represent the standard deviation of the corresponding values calculated from three independent samples. **e** The cycling test of the same CdPS$_3$ _25 electrode for HMF hydrogenation at −0.7 V$_{RHE}$. **f** The retention concentration of HMF, accumulated BHMF (bottom), and the corresponding chronoamperometric test over passed charge (top) during HMF hydrogenation at −0.7 V$_{RHE}$.

and presence of 20 mM HMF. As shown in Fig. 3a, an obvious cathodic peak is observed at ~0.65 V versus reversible hydrogen electrode (V$_{RHE}$) upon the addition of 20 mM HMF, which suggests that CdPS$_3$ is active in catalyzing the HMF hydrogenation reaction. Notably, the potential required for a current density of 10 mA cm$^{-2}$ is reduced by about 120 mV when 20 mM HMF is introduced to the PBS electrolyte (Supplementary Fig. 6). Previous seminal works have unveiled that different factors such as the nature of electrolyte and organic additives may contribute for changing the surface property and catalytic activity of electrocatalysts[41,42]. Here, we systematically studied the role of the electrolyte and HMF in activating the catalyst surface during the electrochemical conversion of HMF to BHMF. We firstly activated the CdPS$_3$ nanosheet in 0.1 M PBS with and without the HMF molecule via ten cyclic voltammetry (CV) cycles, and these electrodes were directly utilized to electro-catalyze hydrogenation of HMF (10 mM) to BHMF at −0.6 V$_{RHE}$. We found that the electrode activated in the presence of HMF (0.1 M PBS & 10 mM HMF) demonstrated nearly 1.7 times higher FE for BHMF synthesis at −0.6 V$_{RHE}$ than that activated in only 0.1 M PBS (46.5% vs 27.5%, Fig. 3b). This result indicates HMF molecule in PBS have a positive effect on activating and enhancing the activity of CdPS$_3$ electrode. To further explore the effect of activation cycles on the performance of the electrocatalyst, we collected the FE and BHMF yield values using the CdPS$_3$ nanosheet electrode activated at 10, 25, and 50 CV cycles (electrodes denoted as CdPS$_3$_10, CdPS$_3$_25, and CdPS$_3$_50), and highest FE of 48.1% is achieved on CdPS$_3$_25 electrode. Subsequently, the CdPS$_3$_25 nanosheet electrodes were used for HMF hydrogenation to BHMF, and 1-h electrolysis was performed at different potentials to achieve the FE and production rate for evaluating its activity (Supplementary Fig. 7). Note that a maximum FE toward BHMF synthesis is achieved when the HMF concentration is 10 mM, while a further increase in HMF initial concentration to 50 mM significantly

reduced FE$_{BHMF}$ (Supplementary Fig. 8). The product BHMF was quantified by nuclear magnetic resonance (NMR, Supplementary Figs. 9, 10). As shown in Fig. 3c, the BHMF production rate gradually increased from 0.63 ± 0.12 to 4.96 ± 0.16 mg/h with change in cathodic potential from −0.55 to −0.7 V$_{RHE}$. However, a further increase in the cathodic potential to −0.75 V$_{RHE}$ results in a decrease in BHMF production rate (4.02 ± 0.2 mg/h), which is probably due to the competing HER that could begin to take the lead at a higher applied potential. Indeed, the hydrogen gas bubbles were hardly seen from the electrode surface until an applied potential equivalent to −0.75 V$_{RHE}$ was utilized. Accordingly, the calculated FE also shows a similar trend and the highest FE reaches 91.3 ± 2.3 % at −0.7 V$_{RHE}$, which is comparable to other reports elsewhere (Table S1). As a comparison, the CdPS$_3$ electrode activated in an electrolyte containing 20 mM HMF in 0.1 M borate buffer solution (BBS) electrolyte was also tested. The FE for BHMF synthesis in BBS is only 59.3 % at −0.7 V$_{RHE}$, which suggests that the electrolyte plays a role during the HMF hydrogenation process. The same measurements conducted on the clean carbon cloth substrate exhibit negligible catalytic activity (Supplementary Figs. 11, 12), further confirming the HMF hydrogenation activity is from the CdPS$_3$ NS catalyst. To intuitively investigate the charge transfer ability in the presence and absence of HMF, we further employ electrochemical impedance spectroscopy (EIS) measurements (Supplementary Fig. 13a–c). The CdPS$_3$_25 electrode exhibits smaller charge transfer resistance (R$_{ct}$) in 0.1 M PBS containing 10 mM HMF (R$_{ct}$ ~ 9.45 Ω, Supplementary Fig. 13b) compared with that in 0.1 M PBS (R$_{ct}$ ~ 15 Ω, Supplementary Fig. 13a).

To assess the stability of our electrocatalyst, we perform the recycling test by applying a constant potential of −0.7 V$_{RHE}$ for 0.5 h per cycle and evaluate the BHMF production rate and corresponding FE values. After each electrolysis cycle, the electrolyte was renewed. As

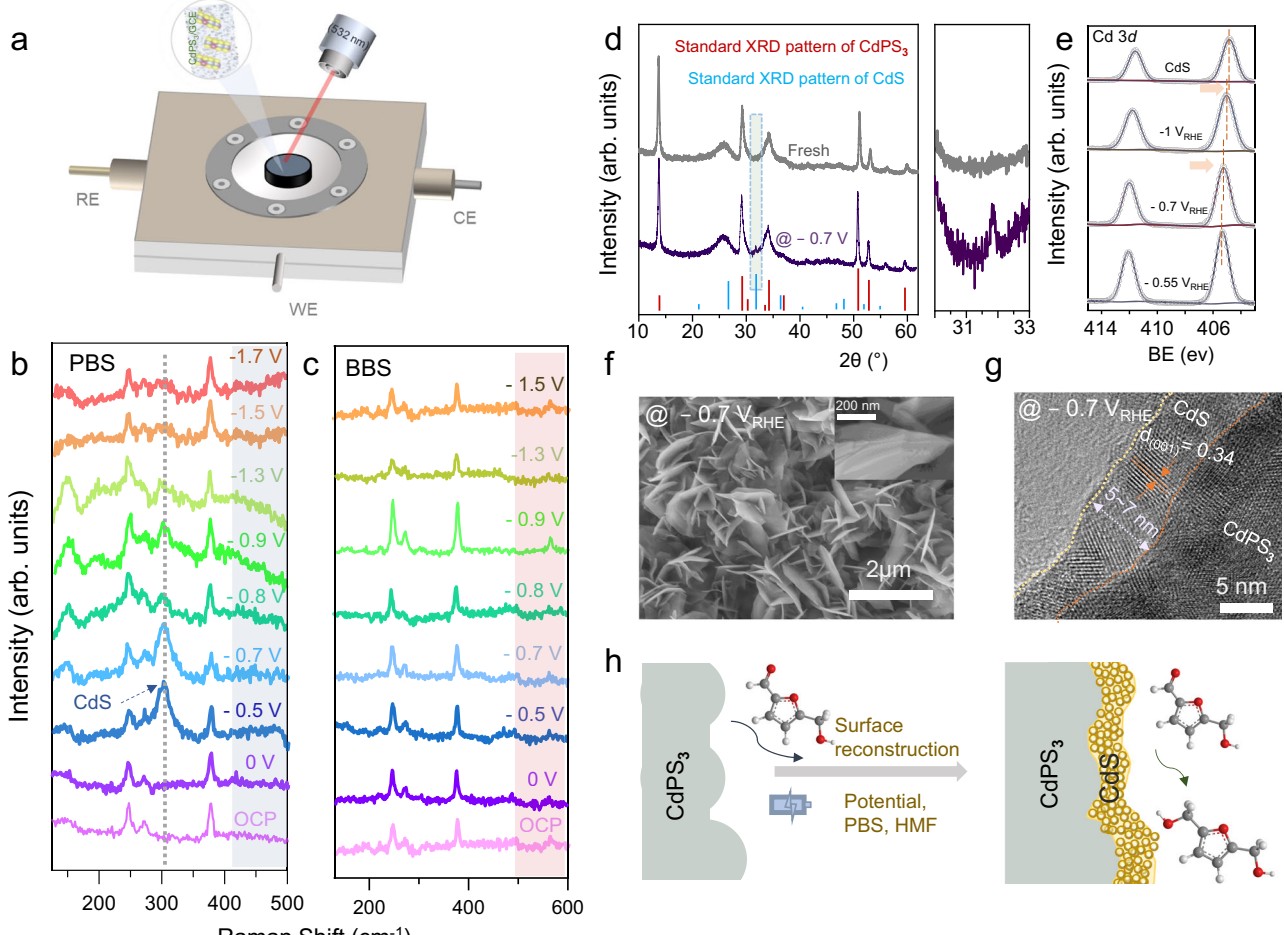

**Fig. 4 | Understanding the origin of electrocatalytic activity via in situ Raman and post-structural characterizations. a** Schematic illustration of the electrochemical cell for in-situ Raman spectroscopy analysis. In situ Raman spectra collected on CdPS$_3$ electrode at the varying potentials in 0.1 M PBS (**b**) and 0.1 M BBS (**c**) electrolytes. **d** Ex-situ XRD characterization of the fresh and spent over electrode after 3 h HMF hydrogenation. **e** XPS spectra of CdS (fresh) and CdPS$_3$_25 electrodes after 3 h of HMF hydrogenation reaction at different applied potentials of −0.55, −0.7, and −1.0 V$_{RHE}$. SEM (**f**), and HRTEM (**g**) characterizations of the CdPS$_3$_25 nanosheet sample after 3 h of HMF hydrogenation reaction. **h** Schematic illustration depicting in situ surface reconstruction of CdPS$_3$ during HMF hydrogenation.

shown in Fig. 3e, the CdPS$_3$_25 nanosheet electrocatalyst demonstrates almost nearly similar FE and BHMF yield for seven continuous cycles, indicating its robust stability to obtain BHMF with high selectivity. Next, we further evaluate the charge-dependent BHMF production and the retained HMF concentration by monitoring the reactant and product concentration after passing 10 C charge each time (Fig. 3f). The BHMF yield and retained HMF concentration could reach 7.6 and 1.56 mM, respectively, after passing 60 C of charge (Supplementary Fig. 14).

**Insight into catalyst structural evolution**

It is well documented that the structurally reconstructed surface of an electrode could provide an opportunity to enhance its catalytic activity via tuning the adsorption energies of reactants and vital intermediate species[17]. This calls for widespread interest in understanding the structural evolution associated with electrochemical conversions. More importantly, a clear picture and an in-depth understanding of the structure and bonding characteristics of an electrode material could give a clue about the reconstruction site of an electrocatalyst. For instance, the strong covalent character of metal-sulfur bond (M-S) than metal–oxygen bond (M-O)[43] endows lattice sulfur to be more reactive than lattice oxygen during catalysis, offering a great possibility for the M-S bond to undergo structural reconstruction[44]. In our case, CdPS$_3$

consists of CdS$_6$ and P$_2$S$_6$ polyhedra units that share edges to form a CdPS$_3$ sheet. Given that the P-S bonds own more covalent character than the Cd-S, a catalyst surface reconstruction could preferably undergo through the P$_2$S$_6$ unit. Recalling the above electrochemical measurements under different CV cycle activation processes, we hypothesize that the reconstructed surface of the CdPS$_3$ nanosheet might have led to a relatively higher efficiency toward HMF hydrogenation to form BHMF.

To track the structural evolution and identify the actual active sites during electrochemical HMF hydrogenation, we employ an in situ Raman spectroscopic technique using a custom-designed electrochemical cell (Fig. 4a and Supplementary Fig. 15). The Raman peaks at 246, 271.5, and 375 cm$^{-1}$ corresponding to the $A_{1g}$ and $E_g$ phonon modes are clearly observed for CdPS$_3$ nanosheet electrode at open circuit potential and 0 V (Fig. 4b). When the applied potential is −0.5 V$_{RHE}$, a new Raman peak at ~301 cm$^{-1}$, assignable to CdS species appears and gradually more discernable at −0.7 V$_{RHE}$ (Supplementary Fig. 16). The intensity of this peak decreases with a change in potential from −0.8 to −1.3 V$_{RHE}$, and finally becomes less obvious at −1.5 V$_{RHE}$. Note that the signal at 301 cm$^{-1}$ of CdS is not obviously observed on two different regions on our operando conditions: (i) when a cathodic potential more positive than the onset potential of HMF reduction is applied, and (ii) when large cathodic potential (more negative than

$-1.5\,V_{RHE}$) is applied. These results suggest that the surface of the CdPS$_3$ nanosheet is reconstructed and CdS species would be generated along with the HMF hydrogenation process. To clearly understand the effect of electrolytes during the HMF hydrogenation, we also performed an in situ Raman spectra analysis by using BBS (0.1 M BBS, pH = 9.2) as electrolyte. As shown in Fig. 4c, only the Raman peaks for CdPS$_3$ appeared under all applied potentials and no peak corresponding to CdS species is seen in the BBS electrolyte. This comparison suggests that the reconstructed surface of CdPS$_3$ in PBS electrolyte is key for improving the HMF hydrogenation efficiency in Fig. 3d. As a comparison, we also examine the catalytic activity of pure CdS sample cast on the same carbon cloth substrate (Supplementary Fig. 17). This electrode could only achieve a BHMF production rate and FE of ~2.2 mg/h and 44.8 %, respectively, at $-0.65\,V_{RHE}$ (Supplementary Fig. 18).

We propose that the P-P bond in the P$_2$S$_6$ unit of CdPS$_3$ could likely undergo structural change during the electrochemical HMF hydrogenation process in 0.1 M PBS electrolyte. Hence, we anticipate that the slightly stronger Cd-S bonds are retained while the change in the P-P bond microenvironment could have led to the advent of CdS on the electrode surface. To substantially evidence this, we employ comprehensive ex-situ morphological and structural characterizations on the spent over CdPS$_3$ electrocatalyst. The ex-situ XRD was first performed to trace any change in the crystalline structure of our electrode after HMF hydrogenation electrocatalysis. As shown in Fig. 4d, a new peak located at 31.8 degree corresponding to the (110) facet of a cubic phase CdS is detected, indicating the formation of crystalline CdS on the CdPS$_3$ nanosheet during electrocatalysis. Furthermore, we also employ ex-situ XPS characterization after a 3 h HMF hydrogenation test at different applied potentials. By comparing the binding energy of Cd $3d_{5/2}$ of pure CdPS$_3$ (405.8 eV) and CdS (404.85) with those after 3 h HMF hydrogenation test at different potentials, a negative shift is observed after cathodic potential of $-0.55\,V_{RHE}$ (405.35 eV), $-0.7\,V_{RHE}$ (405.21 eV), $-1.0\,V_{RHE}$ (405.05 eV) are applied (Fig. 4e). The result unambiguously suggests that the surface of CdPS$_3$ electrode is in situ transformed to yield an active surface containing CdS during applying potentials. We then conducted postmortem morphological characterization on the CdPS$_3$ nanosheet electrode. From the SEM image of the spent over electrocatalyst (Fig. 4f and Supplementary Fig. 19), we noticed no apparent morphological change. The TEM and HR-TEM images (Fig. 4g and Supplementary Fig. 19b) collected from the CdPS$_3$ nanosheets after 3h-HMF hydrogenation at $-0.7\,V_{RHE}$ shows an obvious construction layer of 5–7 nm that possesses a lattice fringe of 0.34 nm on the new reconstructed layer of the nanosheet, which can be indexed to (001) facet of cubic phase CdS. Together with the in-situ Raman, ex situ XRD, and XPS characterizations, the clearly observed reconstructed surface in the HR-TEM image solidly confirms the formation of CdPS$_3$/CdS heterostructure, which is the possible active site for efficient HMF hydrogenation (Fig. 4h).

To study whether the presence of other furan-based compounds could give rise to CdPS$_3$ surface reconstruction under similar activation process in Fig. 3b, we also examined the catalytic activity of furfural reduction on CdPS$_3$ electrode in 0.1 M PBS. As shown in the LSV curves (Supplementary Fig. 20a), the CdPS$_3$ electrode is active towards furfural hydrogenation. Next, we examine the furfural hydrogenation catalytic performance on CdPS$_3$ electrodes activated for 10 CV cycles in only PBS, 10 CV cycles in PBS containing 10 mM HMF, and 25 cycles in PBS and 10 mM HMF. It is interesting to note that the catalyst activated via 25 CV cycles in 0.1 M PBS and 10 mM HMF display relatively higher FE of ~63.6 % at $-0.6\,V_{RHE}$ (Supplementary Fig. 20b, c), which clearly shows the pivotal role of furfural in catalyst activation. Meanwhile, similar to that exhibited in HMF hydrogenation process, the post structural characterization of the spent electrode after furfural hydrogenation clearly depict the formation of CdPS$_3$/CdS

heterostructure (Supplementary Fig. 21). Next, we established a control experiment by employing two-dimensional In$_2$S$_3$ and CdPSe$_3$ nanosheets (Supplementary Figs. 22–25). The HMF hydrogenation performance of these electrocatalysts was evaluated after activating them under different CV cycles. Unlike the CdPS$_3$ electrode, both In$_2$S$_3$ and CdPSe$_3$ nanosheets did not show appreciable change in the catalytic performance under different CV activation process. From the post-reaction characterizations, there is no change in the catalyst structure and surface after prolonged operation, which evidence the absence of surface reconstruction during HMF hydrogenation process on In$_2$S$_3$ and CdPSe$_3$ nanosheet electrode. The control experiments clearly show that the specific reconfiguration of CdPS$_3$ is the main attributing factor to the promoted performance for hydrogenation of HMF.

To further understand the role of the in situ generated CdS layer, we also synthesized a heterostructure through directly depositing CdS quantum dots on CdPS$_3$ nanosheets (CdPS$_3$/CdS$_{QD}$) using one-step chemical bath deposition method (Supplementary Fig. 26, See details in the experimental section). Interestingly, the as-fabricated CdPS$_3$/CdS$_{QD}$ electrocatalyst demonstrate a FE$_{BHMF}$ of 75.9% at $-0.7\,V_{RHE}$ (Supplementary Fig. 27a, b), which is better than the pristine CdS (maximum FE$_{BHMF}$ = 44.8 % at $-0.65\,V_{RHE}$) but lower than that of the in situ generated CdPS$_3$/CdS heterostructure electrode (CdPS$_3$_25, FE$_{BHMF}$ = 91.3%). These results further demonstrate the crucial role of the in situ generated heterointerface for efficient HMF hydrogenation.

## Mechanism for electrocatalytic HMF hydrogenation

To shed light on the role of in-situ generated CdPS$_3$/CdS heterointerface for HMF hydrogenation and understand the underlying reaction mechanism, we investigate the energy profiles of HMF hydrogenation and HER, using DFT calculations. Generally, the electrochemical conversion of HMF to BHMF on the cathode surface may follow either Langmuir Henshelwood (LH, proton adsorbed on the catalyst as a source) or Eley Rideal (ER, proton from electrolyte) mechanisms depending on the nature of electrode and reaction conditions[9,45]. For the LH mechanism, H* and HMF* species adsorbed on the electrode surface react to form the hydrogenated product. Whereas, a net H-atom addition to HMF would be carried via a direct transfer of H$^+$ from the electrolyte and electron from the cathode to yield the product on the electrode surface in the case of the ER reaction mechanism[9]. Our electrolytic HMF hydrogenation was carried out under a mild alkaline environment (0.1 M PBS, pH = 9.2). Hence, we reasoned that HMF hydrogenation over CdPS$_3$/CdS heterointerface would follow the LH mechanism. To further explore this, we evaluate the CV curves of the CdPS$_3$_25 electrode in electrolyte without and with continuous HMF addition. As shown in Fig. 5a, a peak at 0.065 $V_{RHE}$, which could be attributed to H* desorption from the electrode surface[46,47], is observed in the anodic scan when pure electrolyte (0.1 M PBS) is utilized. Upon the addition of HMF with varying concentrations (1–50 mM), the H* desorption peak momentously decreased and completely inhibited when the HMF concentration reached 50 mM. As the integral area of these adsorption peaks is related to the H* coverage on the active site[48], the minimized peak area upon HMF addition could be mainly attributed to the consumption of proton by adsorbed HMF molecule on the catalyst surface.

Next, the CdPS$_3$(001), CdS(110) slabs, and CdPS$_3$(001)/CdS(110) heterointerface models are built based on our experimental results and the corresponding Gibbs free energies for H* ($\Delta G_{H^*}$) and HMF* adsorptions ($\Delta G_{HMF^*}$) are calculated (see Methods part for the details). The pristine CdPS$_3$(001) surface exhibits much higher $\Delta G_{H^*}$ values of 1.95 and 1.40 eV for the P and S sites, respectively, which suggest their weak affinity to bind the first H* adatom. Moreover, the H* adsorption on the Cd sites of both CdS(110) and CdPS$_3$/CdS(110) models would be an uphill process with a higher energy barrier of 1.73 and 1.28 eV, respectively. Interestingly, the S site at the interface of the CdPS$_3$/

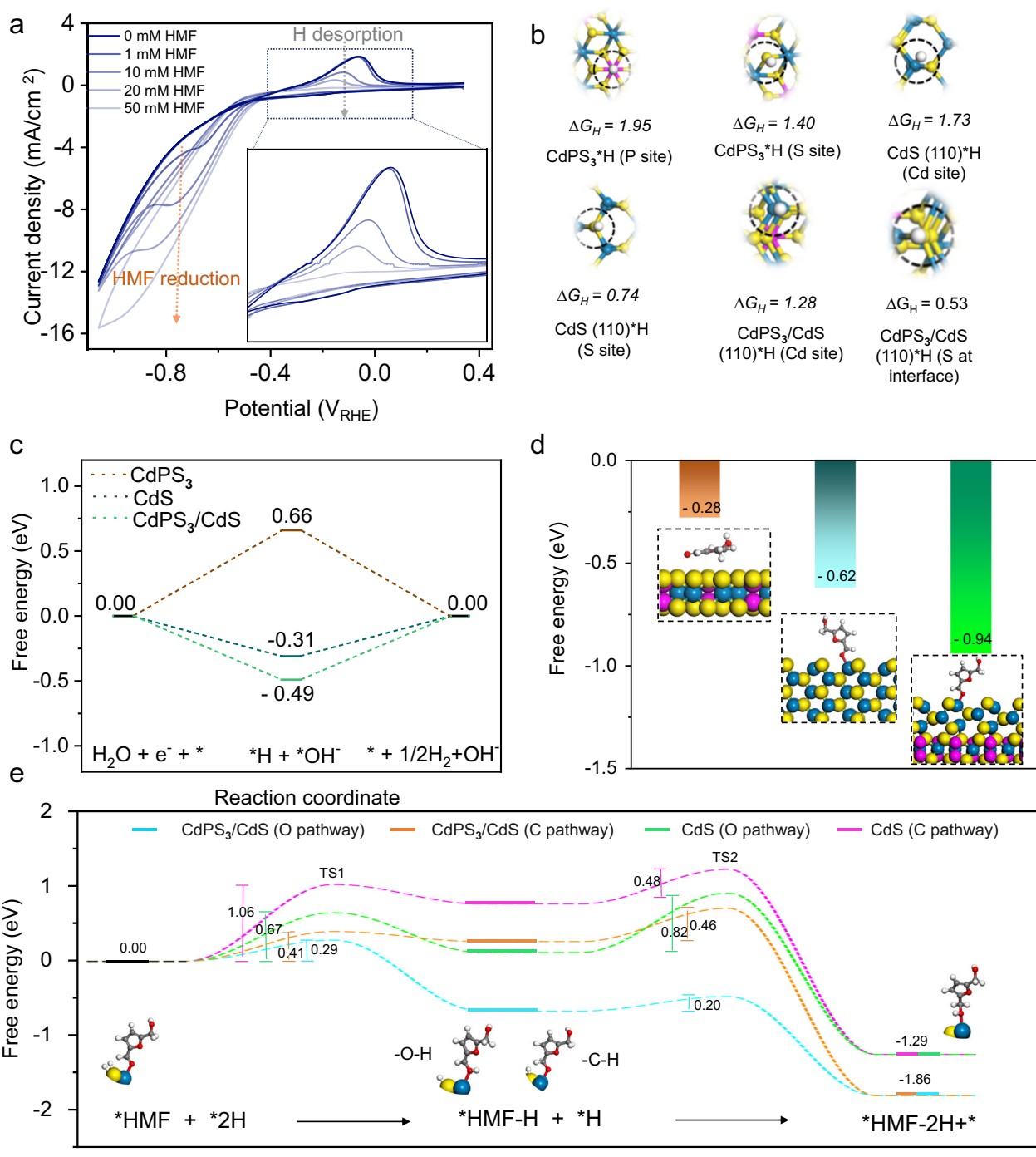

**Fig. 5 | Theoretical calculations for the reaction pathway and mechanism on in situ generated CdPS₃/CdS heterostructure for HMF hydrogenation to BHMF.**
**a** The cyclic voltammogram of CdPS₃_25 sample with the addition of varying HMF concentration. In set is the H* desorption peak. **b** Calculated free energy of H* adsorption on CdPS₃, CdS, and CdPS₃/CdS interface. **c** The calculated ΔG value for water dissociation to give an adsorbed H* species on CdPS₃, CdS, and CdPS₃/CdS heterostructure. **d** Free energy of HMF adsorption on CdPS₃, CdS, and CdPS₃/CdS interface. **e** Calculated free energy values for HMF hydrogenation to BHMF at CdS and CdPS₃/CdS interface using the Langmuir-Hinshelwood (LH) reaction mechanism. The free energies of HMF hydrogenation were computed considering both O- and C- pathways.

CdS(110) heterostructure exhibits a relatively smaller $\Delta G_{H^*}$ of 0.53 eV, indicating the crucial role of heterointerface formation to afford S active site for H* adsorption (Fig. 5b). It is worth noting that the electrode surface should enable the facile activation and dissociation of water to form adsorbed H* species on its surface in the case of LH mechanism. Hence, we also evaluate the Gibbs free energy change for the formation of H* from $H_2O$ dissociation, as shown in Fig. 5c. The

formation of H* on CdPS₃/CdS(110) heterostructure is found to be an exothermic process with a relatively more negative ΔG of −0.49 eV compared with CdS (−0.31 eV); while pure CdPS₃ requires overcoming an energy barrier of +0.66 eV. In addition, we calculated the $\Delta G_{HMF^*}$ (Fig. 5d) and found that HMF prefers to adsorb on the Cd site of CdPS₃/CdS heterostructure with $\Delta G_{HMF^*}$ of −0.94 eV; while it is weakly adsorbed on either the pristine CdPS₃ (−0.28 eV) or pure CdS surface

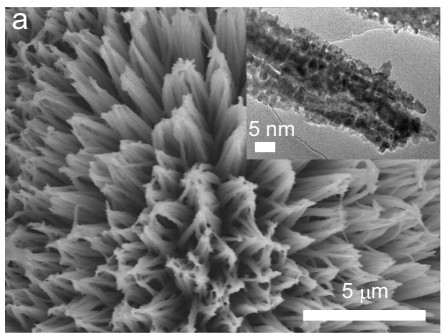

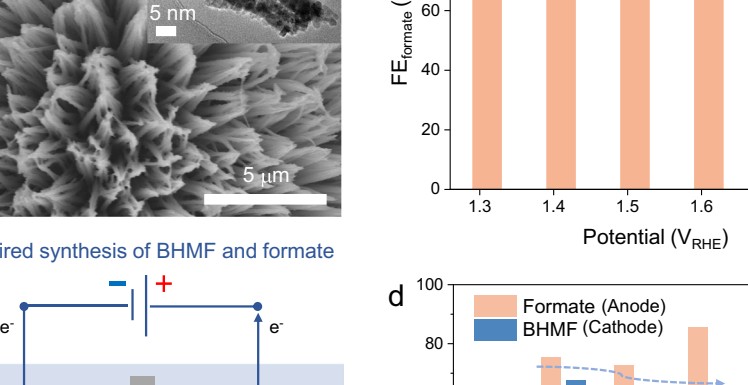

Paired synthesis of BHMF and formate

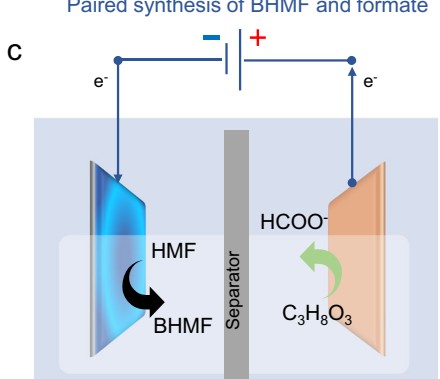

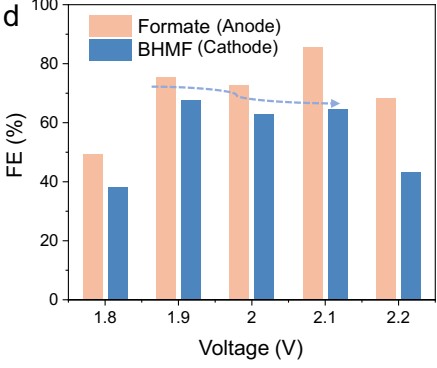

**Fig. 6 | Paired electrochemical synthesis of BHMF and formate through HMF hydrogenation and glycerol oxidation in a two-electrode system. a** SEM image of $MnCo_2O_{4.5}$ nanorod. Inset is the corresponding TEM image. **b** potential dependent formate production from glycerol oxidation over $MnCo_2O_{4.5}$/NF foam anode. Error bars represent the standard deviation of the corresponding values calculated from the measurement of three independent samples. **c** Schematics depicting concurrent production of BHMF and formate at cathode and anode, respectively. **d** Electrochemical synthesis of BHMF and formate from HMF hydrogenation and glycerol oxidation using CdPS$_3$_25 and $MnCo_2O_{4.5}$/NF as cathode and anode, respectively.

(−0.62 eV). The enhanced HMF adsorption on CdPS$_3$/CdS heterostructure could be attributed to the altered electronic structure, as suggested by the projected density of states (PDOSs) calculation (Supplementary Fig. 28). The occupied 4$d$ orbital of the Cd site would be much closer to the Fermi level, leading to a stronger adsorption of HMF molecular on the Cd sites. The aforementioned calculation results suggest that HMF* and H* adsorption on CdPS$_3$(001)/CdS(110) heterostructure takes place at different active sites of Cd and S at the interface.

From the aforementioned experimental and theoretical evidence, HMF hydrogenation is proposed to follow the LH reaction mechanism. This process could proceed via initial hydrogenation of either C- or O-site of the carbonyl (C=O) functional group of HMF to obtain the *H-HMF intermediate or BHMF molecule (i.e., H2-HMF). The calculated energy barrier for each step of HMF conversion to BHMF in LH mechanisms on both CdS and CdPS$_3$(001)/CdS(110) models is lower for the O-pathway than that of the C-pathway (Fig. 5e). The first hydrogenation step (HMF* + *H → TS1 → H·HMF*) on CdPS$_3$(001)/CdS(110) heterostructure under O-pathway is an energetically uphill process with ΔG of only 0.29 eV, which is significantly lower than that on CdS (0.67 eV). Thanks to the heterointerface, the second step of protonation (H·HMF* + H* → TS2 → H2-HMF*) on CdPS$_3$(001)/CdS(110) surface requires a smaller energy barrier of 0.20 eV compared with that on CdS (0.82 eV), implying that HMF hydrogenation could be energetically more favored on CdPS$_3$(001)/CdS(110) heterostructure. Importantly, the rate determining step (RDS) for CdPS$_3$(001)/CdS(110) models would be the first hydrogen transfer process (HMF* + *H → TS1 → H·HMF*). Whereas, the RDS for the CdS model could be the second hydrogen transfer process (H·HMF* + H* → TS2 → H2-HMF*). Note that our DFT calculation also suggests that the HMF hydrogenation under the ER mechanism could follow the O-pathway

(Supplementary Fig. 29), with a small energy barrier on CdPS$_3$(001)/CdS(110) heterostructure (0.14 eV) than CdS(110) model (0.52), further revealing the crucial role of as-formed heterointerface for BHMF synthesis through HMF hydrogenation.

In short, the combined theoretical and experimental studies show that HMF hydrogenation on CdPS$_3$(001)/CdS(110) could proceed via the LH mechanism following the O-pathway, and the surface-bounded CdS plays a crucial role in manipulating the adsorption energetics of H* and HMF*. The in-situ generated interface could simultaneously enhance the HMF adsorption on Cd sites and H adsorption on S sites nearby to complete the hydrogenation process with a lower energy barrier. Given that HER is one major competing reaction to HMF, the less favorable recombination of adsorbed proton to give H$_2$ at mild electrode potential further supports the higher selectivity towards HMF on the CdPS$_3$(001)/CdS(110) catalyst.

Inspired by the catalytic activity of our electrocatalyst, we further construct a coupled electrochemical synthesis system via pairing HMF hydrogenation reaction on the cathode with glycerol oxidation reaction (GOR) on the anode to replace the less-valued water oxidation. Taking advantage of the previous works that uncover the promising catalytic activity and performance of spinel oxides, we choose $MnCo_2O_{4.5}$ nanorod arrays on nickel foam (i.e., $MnCo_2O_{4.5}$/NF) as an anode (Fig. 6a and Supplementary Figs. 30–32, and Supplementary Note II). First, the GOR over $MnCo_2O_{4.5}$ anode was evaluated in a three-electrode system H-type cell. The preferential oxidation of glycerol over water is clearly seen from the polarization curve, which is due to the thermodynamically more favorable GOR than OER (Supplementary Fig. 33). From the EIS measurement in Supplementary Fig. 34, the $MnCo_2O_{4.5}$/NF electrode exhibit lower R$_{ct}$ when tested in the presence of 0.1 M glycerol (5.7 Ω) than in only 1 M KOH (26.1 Ω), which indicates faster electron transfer during the GOR than OER. The potential

dependent GOR performance of $MnCo_2O_{4.5}$ anode was investigated in a three-electrode system through chronoamperometric measurement after passing a fixed charge of 20 C. As shown in Fig. 6b, the FE for formate ($FE_{formate}$) could reach nearly 90.6 % at an applied potential of 1.6 $V_{RHE}$. Upon pairing the $CdPS_3$ as cathode for HMF hydrogenation and the as-synthesized $MnCo_2O_{4.5}$ electrode as the anode, FEs of 67.7% and 75.25 % for BHMF and formate synthesis, respectively, are obtained at 1.9 V (Fig. 6c, d). Note that the FE results obtained in the paired electrochemical process are lower when compared with the individual electrochemical process (HMF paired with OER and GOR paired with HER), which further deserve examining alternative anodic oxidation reaction.

## Discussion

In summary, we have presented a space-confined chemical vapor conversion (SCCVC) strategy to synthesize $CdPS_3$ NS electrocatalyst capable of catalyzing electrochemical HMF hydrogenation to BHMF. In situ and ex situ structural characterizations indicate that the $CdPS_3$ catalyst undergo surface reconstruction leading to the formation of $CdPS_3$/CdS heterostructure. The in situ Raman spectroscopy analysis revealed that $CdPS_3$ electrode could undergo surface transformation in 0.1 M PBS electrolyte when a cathodic potential above the onset of HMF reduction potential is applied, while no apparent change in the material structure is seen in 0.1 M BBS electrolyte. Interestingly, the obtained $CdPS_3$/CdS heterointerface showed high FE and BHMF yield of 91.3 ± 2.3 % and 4.96 ± 0.16 mg/h, respectively, at −0.7 $V_{RHE}$. Theoretical calculations show that the in situ generated $CdPS_3$/CdS heterostructure enables the optimal adsorption of HMF* and H* at Cd and S active sites, respectively, and facilitates the hydrogenation steps by minimizing the energy barrier at each step through the LH reaction mechanism. When paring the HMF hydrogenation reaction on the $CdPS_3$ cathode with the glycerol oxidation reaction on the $MnCo_2O_{4.5}$ anode, two high-valued chemicals of BHMF and formate could be concurrently generated with high selectivity (>65%) under the cell voltage of 1.9 V. The results described herein not only disclose an efficient electrocatalyst for biomass hydrogenation but also offer a fundamental understanding of the structure-activity relationship, which could facilitate the rational design of novel phosphorus and sulfur-rich compounds for energy conversion applications.

## Methods

### Synthesis of $CdPS_3$ catalyst

The $CdPS_3$ nanosheets were synthesized using a SCCVC method utilizing the CdS on the carbon cloth (i.e., CdS/CC) as a precursor in a two-zone tube furnace. Typically, a powder mixture (0.75 g) containing P and S (1:3 ratio) was placed at the front zone, and the as-prepared CdS/CC was placed at the back zone. Then, the furnace was pumped with Ar gas to create a vacuum environment. Subsequently, the front and back zones were simultaneously heated to 280 °C and 420 °C, respectively, within 20 min under 100 sccm Ar gas flow. The reaction lasted 20 min, and the final temperature of the front and back zone was 300 and 420 °C, respectively. The as-obtained $CdPS_3$ NS was collected after naturally cooling the furnace to room temperature.

### Synthesis of $CdPS_3$/$CdS_{QD}$ heterostructure catalyst

To obtain the $CdPS_3$/$CdS_{QD}$ heterostructure catalyst, the CdS quantum dots were directly deposited on the as-prepared $CdPS_3$ nanosheets by a chemical bath deposition method[49]. In a typical synthesis procedure, four beakers are first prepared; the first two contain 0.01 M $Cd(NO_3)_2$ and 0.01 M $Na_2S$ (50 mL each), respectively, and the other two contain distilled water for rinsing the samples. Then, the $CdPS_3$/CC was sequentially immersed in the four beakers for 15 s. The process was repeated four times to obtain CdS quantum dots decorated $CdPS_3$ nanosheets (i.e., $CdPS_3$/$CdS_{QD}$).

## Characterization methods

The morphologies of the as-synthesized electrocatalysts were examined using Scanning electron microscopy (SEM, Hitachi SU8220) and Transmission electron microscopy (TEM, JEM 2100, JEOL Ltd, Japan), which are equipped with EDX. X-ray diffraction (XRD, D/MAX-TTRIII(CBO)) patterns were collected using Cu-Kα radiation ($\lambda$ = 1.5418 Å). Raman spectroscopy measurements were performed at room temperature using an inVia Renishaw system with a 532 nm excitation laser. The Thermo Scientific ESCALab 250Xi that utilized 200 W monochromatic Al Kα radiation was used to acquire the X-ray photoelectron spectroscopy (XPS) spectra of the as-synthesized samples. The C1s peak positioned at 284.8 eV was taken as a reference to correct the binding energies. The analysis chamber was maintained at a base pressure of $3 \times 10^{-9}$ mbar. The atomic force microscopy (AFM, MFP-3D Infinity) was utilized to analyze the thickness of the as-synthesized $CdPS_3$ nanosheets, and the nanosheets were dry-transferred from carbon cloth directly onto $SiO_2$/Si substrate.

## Electrochemical measurements

The electrochemical measurements were carried out in a customized three-electrode cell (H-type glass cell separated by Nafion 117 membrane) using a Princeton Applied Research Potentiostat/Galvanostat (VersaSTAT3) electrochemical analyzer, where Ag/AgCl (saturated KCl), Pt wire, and $CdPS_3$/CC were used as a reference, counter, and working electrodes, respectively. The mass loading of the $CdPS_3$ electrocatalyst was ~1.1 mg/cm$^2$, and it was determined by measuring the weight of the carbon cloth substrate before and after the SCCVC process. Unless specifically mentioned, all the geometric dimensions of the electrodes in the electrochemical tests were 1 × 1 cm$^2$. All potentials reported in this work are quoted with respect to the reversible hydrogen electrode (RHE) and calibrated using the equation $E(V_{RHE}) = E_{Ag/AgCl} + 0.197 + 0.0591 \times pH$. The calibration of the reference electrode was conducted utilizing a three-electrode system in 0.5 M $H_2SO_4$ electrolyte saturated with high-purity hydrogen, employing a cyclic voltammetry method with a scan rate of 1 mV s$^{-1}$. Prior to the electrochemical measurements, the Nafion 117 membrane was treated at 80 °C using 0.5 M $H_2SO_4$, 3% $H_2O_2$, and deionized water for 30 min. The electrochemical reduction of HMF was carried out in 0.1 M phosphate buffer solution (PBS, pH = 9.2 ± 0.01) containing 10 mM HMF, and the total volume of electrolyte in both anode and cathode chamber was 30 ml. By employing EIS at frequencies ranging from 10 Hz to 100 kHz, the solution resistance in our standard three-electrode system was determined to be 6.43 ± 0.15 ohms.

## Calculation of the FE and yield rate of BHMF

The Faradaic efficiency for BHMF formation under different potentials was calculated according to the equation:

$$FE = \frac{\text{mole of BHMF}}{\text{total charge passed}/(F \times 2)} \times 100\% \quad (1)$$

The BHMF production rate was calculated using the following equation

$$r_{BHMF} = \frac{(C_{BHMF} \times V_{BHMF})}{t} \quad (2)$$

The FE for the furfuryl alcohol was also calculated via employing the above equation except using the moles of furfuryl alcohol instead of BHMF.

## NMR determination of BHMF

The product BHMF and furfuryl alcohol was quantified using $^1$H nuclear magnetic resonance ($^1$H NMR) spectra collected on a Bruker AVANCE III (400 MHz). A solution containing both $D_2O$ and dimethyl

sulfoxide (5 mM DMSO) was used as the internal standard. The standard calibration curves for HMF, BHMF, and furfuryl alcohol were established from highly pure commercial samples.

**Electrochemical in situ Raman measurement.** The aforementioned electrochemical cell (Fig. 4a) was used for the in situ Raman spectroscopy analysis. As for the working electrode, the catalyst ink was prepared by sonicating a $1 \times 2$ cm$^2$ CdPS$_3$/CC sample in a mixture of ethanol and water (3:1 ratio, 3 ml) and subsequently cast on a glassy carbon electrode. The platinum wire and Ag/AgCl electrodes were used as counter and reference electrodes, respectively.

**Theoretical calculations and modeling.** The theoretical calculations were carried out by the Vienna ab initio simulation package (VASP) based on the DFT[50]. The spin-polarized DFT calculations were conducted using the Perdew-Burke-Ernzerhof exchange-correlation functional in the VASP. The projector augmented wave method (PAW)[51,52] was utilized with a plane-wave kinetic energy cutoff of 500 eV and a Gaussian smearing of 0.02 eV. To sample the Brillouin zone, we used a $3 \times 3 \times 1$ K-point. The CdPS$_3$, CdS(110), and CdPS$_3$/CdS(110) heterostructure slabs were modeled by a supercell with dimensions of 12.507 Å × 21.662 Å, 11.620 Å × 8.216 Å, and 12.063 Å × 21.101 Å, respectively (Supplementary Data 1). A vacuum gap of 15 Å was added to separate the periodic images of the slab in the direction perpendicular to the surface[53]. During geometry optimization, all atoms were allowed to relax, and the atomic positions were optimized until the forces were less than 0.02 eV/Å. We modeled the effects of van der Waals corrections using Grimme's method with Becke–Johnson damping[54,55]. The transition states (TS) were searched by the Dimer method and further confirmed by vibrational frequency analysis[56]. Only one imaginary frequency was found for each of the TS structures reported in this work.

The free energy difference for all of the elementary steps that involve an electron transfer is calculated by the equation $\Delta G = \Delta E + \Delta ZPE - T\Delta S + \Delta G_U + \Delta G_{pH}$, where $\Delta E$, $\Delta ZPE$, and $\Delta S$ correspond to the energy difference between adsorption energy, zero-point energy, and entropy, respectively. The adsorption energies $\Delta E$ were measured by using DFT. The $\Delta ZPE$ and $T\Delta S$ values were obtained from harmonic vibrational frequency calculations and DFT. $\Delta G_U = -eU$, where U represents a potential based on a standard hydrogen electrode. $\Delta G_{pH}$ represents the Gibbs-free energy correction of the pH, noting that we consider pH = 0 in our computational investigation.

## Data availability
The data generated in this study are included in the paper and its Supplementary Information, and can be obtained from the corresponding authors upon request.

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

## Acknowledgements

H.D. and X.S. acknowledge the support from the National Natural Science Foundation of China (21935001). F.W. acknowledges the support from the National Natural Science Foundation of China (22179029) and Fundamental Research Funds for the Central Universities (buctrc202324). H.D. also acknowledges the support from the National Natural Science Foundation of China (22325805) and Beijing Natural Science Foundation (Grant No. JQ22003). F.W. also acknowledges the support from Young Elite Scientists Sponsorship Program by CAST (2023QNRC001), the Young Elite Scientists Sponsorship Program by BAST (BYESS2023093), and the Youth Innovation Promotion Association CAS.

## Author contributions

F.W., H.D., and X.S. supervised the project. M.G.S and F.W conceived and designed the project. M.G.S perform the experiment. K.H. carried out the theoretical calculation. F.T.D, B.L.W, S.H, N.G, X.Z, Y.Y, Z.W, C.C, and W.L. assisted in material characterizations. M.G.S., F.W., H.D., and X.S. wrote the manuscript in consultation with all the other authors.

## Competing interests

The authors declare no competing interests.
