## [Peer Review File · Nature Communications]

Deciphering In-Situ Surface Reconstruction in Two-dimensional CdPS₃ Nanosheets for Efficient Biomass HydrogenationREVIEWER COMMENTS

Reviewer #1 (Remarks to the Author):

In this manuscript, the authors reported the synthesis of a CdPS3 nanosheet electrocatalyst by a space-confined chemical vapor conversion strategy and its application in the electrochemical hydrogenation of 5-hydroxymethylfurfural (HMF) to 2,5-bis(hydroxymethyl)furan (BHMF). They first discussed the synthesis approach, physical characterizations and electrochemical properties of the synthesized CdPS3 catalyst in sequence. Subsequently, they systematically investigated the surface reconstruction process of CdPS3 to CdPS3/CdS heterostructure using in situ (Raman spectroscopy) and ex situ (XRD, XPS, and HR-TEM) structural characterization techniques, which makes the submitted manuscript interesting and may be of great interest to many researchers in the electrochemical community. The density functional theory calculations have also been employed to gain insight into the CdPS3/CdS heterointerface and its underlying electrocatalytic mechanism for electrocatalyzing HMF hydrogenation. As illustrated in the manuscript, the in situ generated CdPS3/CdS heterostructure could modulate the electronic structure of Cd and S active sites to optimize their adsorption of HMF* and H*, respectively. Overall, this work is interesting and might open up an avenue to design and fabricate efficient electrocatalytic materials. However, the following several issues need to be addressed before it can be accepted for publication.

1. To study the effect of activation cycles on electrocatalytic performance, CdPS3 catalysts in this work were activated with different CV cycles (10, 25, and 50). As a result, the CdPS3 activated with 25 CV cycles (CdPS3_25) showed the best performance. What are the underlying reasons behind the superior performance of CdPS3_25. More characterization results and/or discussion should be provided to figure it out.
2. As shown in Figure 4g, the thickness of CdS in CdPS3/CdS heterostructure is about 5-7 nm. Whether the authors considered that the thickness of CdS in CdPS3/CdS heterostructure may significantly affect its electrocatalytic activity. Researchers in related fields would like to see a high-quality study that cover this important information.
3. The authors claimed on page 15 that the potential determining step (PDS) for both CdS and CdPS3(001)/CdS(110) models is the first hydrogen transfer process (i.e., $\text{HMF}^* + \text{H}^+ \rightarrow \text{TS1} \rightarrow \text{H-HMF}^*$). However, this conclusion cannot be drawn from Figure 5 on page 16.
4. Since the CdPS3 catalyst after surface reconstruction (CdPS3/CdS heterostructure) displays excellent electrocatalytic properties, have the authors considered or tried to directly synthesize catalysts with such core@shell heterostructure to verify this reasonable hypothesis? If the authors can provide relevant data in this aspect, it may significantly improve the quality of the work. Below some additional minor points also should be addressed.
 1. The abbreviations (i.e., BHMF and HMF) in the Abstract should be defined when they first appear. Similar minor errors can also be found on Page 4 (FE) of the main text.
 2. There are many typing errors on page 5 (i.e., Fig. 1a, Figure 1b), page 6 (i.e., Fig. 1d,e, Fig. 1f, Fig. 1g.,h) needed to be corrected.
 3. The authors missed several seminal works on dynamic surface reconstruction of electrocatalysts. e.g., Nat Commun 13, 2207 (2022); Proc. Natl. Acad. Sci. U. S. A., 118, e2023421118 (2021); Nature Mater 16, 925–931 (2017); Chem. Soc. Rev., 50, 8428 (2021). These works are suggested to be discussed in the context.

Reviewer #2 (Remarks to the Author):

Sendeku et al. present an in-depth study on a novel CdPS3 nanosheet electrocatalyst, which readily undergo an in-situ surface reconstruction with the assistance of HMF + electrolyte (PBS). The authors noted that an in situ reconstructed surface of CdPS3 enabled efficient biomass hydrogenation. By utilizing detailed post-reaction characterization and in situ Raman studies, they track the formation of a sub-10 nm CdS layer under the operando regions where HMF hydrogenation takes place. This guarantees the formation of CdPS3/CdS heterointerface, which accounts for the high FE observed in the reaction system. Moreover, density functional theory calculation further highlights the crucial role of in situ generated CdS layer in manipulating the adsorption energies of both HMF* and H*, as well as minimizing the intermediates energy barrier, thus accounting for the efficient HMF hydrogenation process. This work is solid and sound, offering

fresh and inspiring insights that are valuable to the field, as it especially addresses the surface reconstruction phenomena in electrochemistry, which can be used for rational catalyst design not only for HMF hydrogenation but also for other reactions. Therefore, I recommend the acceptance of the manuscript in Nature Communications after considering the following points:

1. Minor points:

- 1) In the introduction the author states that "The rational design of an electrocatalyst for electrochemical conversion of biomass based feedstock requires a full understanding of the reaction process and the key parameters for controlling the electrode surface and interface." please provide a reference for this statement.
- 2) References are missed in the method section for the DFT calculation part. Some related refs should be cited here.
- 3) XRD pattern in Fig . 2b, the broad peak assigned to the carbon cloth should be labeled.
2. In Figure 6, the authors tried to pair the in situ formed CdPS3/CdS heterointerface cathode with MnCo2O4.5/Ni anode to synthesize two valuable chemicals. While detailed structural and electrochemical characterizations are offered for CdPS3/CdS, some characterization for MnCo2O4.5/Ni catalyst (like XPS, EIS) should be provided.
3. When evaluating the performance of the electrode in Fig. 3, 10 mM HMF was utilized. Is this concentration the optimized one? If so, the electrocatalytic performances results related to other HMF concentration should be included.
4. In Figure 3b, the authors found that the prior activation of the CdPS3 electrode in the presence of HMF itself shows better performance for HMF hydrogenation to BHMF. This is interesting. I wonder if the same activation process can also result in some difference in the performance (FE) during the hydrogenation of other furan-based compounds (like furfural to furfuryl alcohol).

Reviewer #3 (Remarks to the Author):

The manuscript presents a novel approach using layered CdPS3 for biomass hydrogenation under considerably milder conditions than typical industrial processes. This study could potentially contribute significantly to the field due to its innovative method and the implications for sustainable energy production.

Strengths:

The concept of utilizing layered CdPS3 for biomass hydrogenation at milder conditions is both innovative and relevant to current research trends in sustainable energy. The manuscript is well-structured, making it easier to understand the process and its potential implications.

Areas for Improvement:

Control Experiments: The central claim that the reconfiguration of CdPS3 leads to the formation of CdS on the surface, which in turn acts as a hydrogenation platform, is intriguing. However, the manuscript currently lacks sufficient control experiments to robustly support this claim. To strengthen the findings:

Consider incorporating control experiments using CdS quantum dots, amorphous surfaces on CdPS3, and other two-dimensional materials like graphene. These controls would help isolate the specific contributions of the CdPS3 reconfiguration to the observed hydrogenation performance.

Clarification of Claims: While the degradation of CdPS3 to form CdS is a central aspect of the study, it would be beneficial to elaborate on how this degradation process directly contributes to the enhanced hydrogenation performance. Additional experimental data supporting this mechanism would add depth to the study.

Conclusion:

The manuscript presents a promising approach to biomass hydrogenation using layered CdPS3. However, to fully substantiate the claims and provide a comprehensive understanding of the

process, additional control experiments are essential. Incorporating these experiments and providing more detailed explanations of the underlying mechanisms would greatly enhance the manuscript's contribution to the field. In its current state, I would recommend a reject-resubmit before considering the manuscript for publication.

Point-by-point reply to the reviewers' comments

Reviewer 1

In this manuscript, the authors reported the synthesis of a CdPS₃ nanosheet electrocatalyst by a space-confined chemical vapor conversion strategy and its application in the electrochemical hydrogenation of 5-hydroxymethylfurfural (HMF) to 2,5-bis(hydroxymethyl)furan (BHMF). They first discussed the synthesis approach, physical characterizations and electrochemical properties of the synthesized CdPS₃ catalyst in sequence. Subsequently, they systematically investigated the surface reconstruction process of CdPS₃ to CdPS₃/CdS heterostructure using in situ (Raman spectroscopy) and ex situ (XRD, XPS, and HR-TEM) structural characterization techniques, which makes the submitted manuscript interesting and may be of great interest to many researchers in the electrochemical community. The density functional theory calculations have also been employed to gain insight into the CdPS₃/CdS heterointerface and its underlying electrocatalytic mechanism for electrocatalyzing HMF hydrogenation. As illustrated in the manuscript, the in situ generated CdPS₃/CdS heterostructure could modulate the electronic structure of Cd and S active sites to optimize their adsorption of HMF* and H*, respectively. Overall, this work is interesting and might open up an avenue to design and fabricate efficient electrocatalytic materials. However, the following several issues need to be addressed before it can be accepted for publication.

Response: We thank the reviewer for the valuable comments and suggestions. We strongly believe that these comments helped us to improve the quality of our manuscript. We addressed the specific comments below:

1. To study the effect of activation cycles on electrocatalytic performance, CdPS₃ catalysts in this work were activated with different CV cycles (10, 25, and 50). As a result, the CdPS₃ activated with 25 CV cycles (CdPS₃_25) showed the best performance. What are the underlying reasons behind the superior performance of CdPS₃_25. More characterization results and/or discussion should be provided to figure it out.

Response: We thank the reviewer for the comment. To understand the surface electronic properties and explore the underlying reason for the optimal catalytic activity on CdPS₃_25 electrode, which is activated in both PBS and HMF, we performed XPS characterizations on the samples activated under different CV cycles. As shown in Fig. R1, the XPS peak of Cd 3d in CdPS₃_25 electrode shifts to lower binding energy compared with the CdPS₃_10 sample. This shift can be associated with the change in electronic property that may emanate from the

activation process, i.e., the formation of stable and optimum CdS layer that could facilitate HMF hydrogenation is achieved when activating the catalyst surface *via* 25 CV cycles. Despite the variation in surface electronic properties of these samples, the difference in the FE within those activation cycles in the presence of 0.1 M PBS and 10 mM HMF is only around 2%, which is negligible. Hence, the key factor to improve the electrocatalytic activity is the formation of the CdPS₃/CdS under CV activation process. The activation CV number will not make a big difference. Considering the slightly better performance on CdPS₃_25, we chose this electrode for further study.

Fig. R1. Comparison of XPS spectra of CdPS₃ electrode activated using different CV cycles in 0.1M PBS electrolyte containing 10 mM HMF.

2. As shown in Figure 4g, the thickness of CdS in CdPS₃/CdS heterostructure is about 5-7 nm. Whether the authors considered that the thickness of CdS in CdPS₃/CdS heterostructure may significantly affect its electrocatalytic activity. Researchers in related fields would like to see a high-quality study that cover this important information.

Response: We agree that the thickness of the CdS layer on CdPS₃/CdS heterostructure may affect its electrocatalytic activity. In this work, we found that the interface between CdPS₃ and CdS is the active sites for HMF hydrogenation. The *in-situ* generated CdS layer with the thickness around 5-7 nm on CdPS₃ can promote the HMF hydrogenation to BHMF in 0.1 M PBS electrolyte. However, rationally controlling the thickness of CdS layer on CdPS₃ is practically challenging. Considering this, we attempt to explore the effect of CdS on the electrocatalytic activity via conducting the HMF hydrogenation test at - 0.7 V_{RHE} over the CdPS₃_25 electrode with different reaction time of 30, 60, and 210. We found that the FE_{BHMF} of CdPS₃ electrode is 86.3, 91.3, and 74.2% when the reaction is carried for 30 min, 60, and 210 min, respectively. Having this at hand, the electrodes are characterized through XPS

analysis and TEM. As shown in Fig. R2, the Cd 3d XPS peak shift towards lower binding energy as the operation time is increased from 30 to 210 min, possibly indicating the formation of more CdS species on CdPS₃. However, the HR-TEM image of the samples run for 30, 60, 180 min still clearly show the CdS layer with almost negligible difference in thickness (5 to 8 nm) on CdPS₃. Hence, the observed change in electronic structure is an attribute of difference in CdS coverage over the entire electrode, leading to some difference in the performance.

In addition, theoretical calculations showed that the Cd of the *in situ* generated CdS and S sites of the CdPS₃/CdS heterostructure are the active sites for adsorption of HMF* and H*, respectively. From this point, it can be understood that a relatively larger or denser reconstructed CdS layer rather than thicker CdS layer on CdPS₃ nanosheet would avail more Cd and S (at interface) active sites that favour the hydrogenation process.

Fig. R2. The time dependent HMF hydrogenation CdPS₃_25 electrode with different reaction time of 30, 60, and 210 min at $-0.7 V_{RHE}$.

Fig. R3. Comparison of XPS spectra of CdPS₃ electrode at $-0.7 V_{RHE}$ after 30 min, 60 min, and 210 min electrocatalytic HMF hydrogenation test.

Fig. R4. HR-TEM image of CdPS₃_25 electrode after 30, 60, 180 min HMF hydrogenation test at $-0.7 V_{RHE}$, demonstrating only slight change in the CdS layer thickness.

3. The authors claimed on page 15 that the potential determining step (PDS) for both CdS and CdPS₃(001)/CdS(110) models is the first hydrogen transfer process (i.e., $HMF^* + *H \rightarrow TS1 \rightarrow H-HMF^*$). However, this conclusion cannot be drawn from Figure 5 on page 16.

Response: We thank the reviewer for pointing out this issue. Based on reviewer's comment, we carefully rechecked Fig. 5e (shown below) and found that the case is not the same for CdS and CdPS₃(001)/CdS(110) models. In the case of CdPS₃(001)/CdS(110) model, the energy barriers needed to overcome the first and second hydrogen transfer process are 0.29 and 0.20 eV, respectively (Fig. 5e). Hence, the first hydrogen transfer process is the rate determining step (RDS). Whereas, for the CdS model, the energy barrier in the second hydrogen transfer process (0.82 eV) is higher than that in the first hydrogen transfer (0.67 eV). Accordingly, we have corrected the description in the revised manuscript. Please see the lines 25-28 on Page 16.

Fig. 5e. Calculated free energy values for HMF hydrogenation to BHMF at CdS and CdPS₃/CdS interface using the Langmuir-Hinshelwood (LH) reaction mechanism. The free energies of HMF hydrogenation were computed considering both O- and C- pathways.

Lines 25-28 on Page 16:

“Importantly, the rate determining step (RDS) for CdPS₃(001)/CdS(110) models is the first hydrogen transfer process ($HMF^* + *H \rightarrow TS1 \rightarrow H-HMF^*$). Whereas, the RDS for the CdS model is the second hydrogen transfer process ($H-HMF^* + H^* \rightarrow TS2 \rightarrow H_2-HMF^*$).”

4. Since the CdPS₃ catalyst after surface reconstruction (CdPS₃/CdS heterostructure) displays excellent electrocatalytic properties, have the authors considered or tried to directly synthesize catalysts with such core@shell heterostructure to verify this reasonable hypothesis? If the authors can provide relevant data in this aspect, it may significantly improve the quality of the work.

Response: We thank the reviewer for the insightful suggestion. We agree with the reviewer that verifying the role of the heterointerface in CdPS₃/CdS could provide further insight to our work. Following the suggestion, we synthesized a heterostructure *via* directly depositing CdS quantum dots (CdS_{QD}) on CdPS₃ nanosheets through one-step chemical bath deposition method to obtain CdPS₃/CdS_{QD} heterostructure (See Supplementary Fig. 26 in the revised manuscript). The XRD pattern of the as-synthesized sample clearly depict the presence of cubic phase CdS (PDF#65-8873) without affecting the CdPS₃ nanosheet (Supplementary Fig. 26a,b). From the HR-TEM images in Supplementary Fig. 26c-d, a lattice fringe of ~ 0.2 and ~ 0.332 nm that corresponds to the (220) and (311) planes, respectively, of CdS is seen, which corroborate well with the XRD patterns in Supplementary Fig. 26a. The Raman spectrum of CdPS₃/CdS_{QD} further present the distinct vibration modes of P₂S₆ unit of CdPS₃ along with the longitudinal vibration mode of CdS (Supplementary Fig. 26b). These characterizations indicate that the CdS_{QD} are successfully decorated on the ultrathin CdPS₃ nanosheet surface. The electrocatalytic HMF hydrogenation performance of the as-fabricated electrocatalyst was then examined in 0.1M PBS electrolyte (Supplementary Fig. 27a,b). Interestingly, the CdPS₃/CdS_{QD} electrocatalyst demonstrated a FE of 75.9%, which is better than the pristine CdS (maximum FE_{BHMF} = 44.8 % at - 0.65 V_{RHE}) but lower than that of the *in situ* generated CdPS₃/CdS heterointerface electrode (91.3%). Besides, the potential dependent FE_{BHMF} of this electrode exhibited similar trend to the CdPS₃-25 electrode in Fig. 3d. These results further highlight the crucial role of the *in situ* generated heterointerface for efficient BHMF synthesis. Following this experimental finding, we have incorporated the related discussion in the revised manuscript. Please see lines 14-23 on Page 13 and lines 25-32 on Page 20 for the details.

Lines 14-23 on Page 13:

“To further understand the role of the in situ generated CdS layer, we also synthesized a heterostructure through directly depositing CdS quantum dots on CdPS₃ nanosheets (CdPS₃/CdS_{QD}) using one-step chemical bath deposition method (Supplementary Fig. 26, See details in the experimental section). Interestingly, the as-fabricated CdPS₃/CdS_{QD} electrocatalyst demonstrate a FE_{BHMF} of 75.9% at 0.7 V_{RHE} (Supplementary Fig. 27a,b), which is better than the pristine CdS (maximum FE_{BHMF} = 44.8 % at $-0.65 V_{RHE}$) but lower than that of the in situ generated CdPS₃/CdS heterointerface electrode (CdPS₃_25, FE_{BHMF} = 91.3%). These results further demonstrate the crucial role of the in situ generated heterointerface for efficient HMF hydrogenation.”

Lines 25-32 on Page 20:

“**Synthesis of CdPS₃/CdS_{QD} heterostructure catalyst.** To obtain the CdPS₃/CdS_{QD} heterostructure catalyst, the CdS quantum dots were directly deposited on the as-prepared CdPS₃ nanosheets by chemical bath deposition method.⁴⁹ In a typical synthesis procedure, four beakers are first prepared; the first two contains 0.01M Cd(NO₃)₂ and 0.01M Na₂S (50 mL each), respectively, and the other two contains distilled water for rinsing the samples. Then, the CdPS₃/CC was sequentially immersed on the four beakers for 15 seconds. The process was repeated four times to obtain CdS quantum dots decorated CdPS₃ nanosheets (i.e., CdPS₃/CdS_{QD}).”

Supplementary Fig. 26. a-b, XRD patterns (a) and Raman spectrum (b) of the as-synthesized CdPS₃/CdS_{QD}. **c-d**, High resolution TEM images of the CdS quantum dots decorated on CdPS₃ nanosheets.

Supplementary Fig. 27. a, LSV curves of CdPS₃/CdS_{QD} electrode tested in 0.1M PBS without and with 20 mM HMF. **b**, Potential dependent electrocatalytic HMF hydrogenation performance of as-synthesized CdPS₃/CdS_{QD} electrode.

Below some additional minor points also should be addressed.

1. The abbreviations (i.e., BHMF and HMF) in the Abstract should be defined when they first appear. Similar minor errors can also be found on Page 4 (FE) of the main text.

Response: We are so thanks for this kind reminder. The mentioned abbreviations are clearly defined when they appear for the first time. Please see page 1 lines 27 and 28 of our revised manuscript for the details in the revised manuscript.

2. There are many typing errors on page 5 (i.e., Fig. 1a, Figure 1b), page 6 (i.e., Fig. 1d,e, Fig. 1f, Fig. 1g.,h) needed to be corrected.

Response: We thank the reviewer for the careful inspection of our manuscript. The aforementioned typos are now corrected. Please see lines 14-15 and 30 on page 5 and lines 9, 13, 15-16 and 18 on page 6 for the details in the revised manuscript.

3. The authors missed several seminal works on dynamic surface reconstruction of electrocatalysts. e.g., Nat Commun 13, 2207 (2022); Proc. Natl. Acad. Sci. U. S. A., 118, e2023421118 (2021); Nature Mater 16, 925–931 (2017); Chem. Soc. Rev., 50, 8428 (2021). These works are suggested to be discussed in the context.

Response: The aforementioned works on dynamic surface reconstruction of an electrode material have been cited and consulted. Please see the reference list (References 13 to 16) for the details in the revised manuscript.

Reviewer 2

Sendeku et al. present an in-depth study on a novel CdPS₃ nanosheet electrocatalyst, which readily undergo an in-situ surface reconstruction with the assistance of HMF + electrolyte (PBS). The authors noted that an in situ reconstructed surface of CdPS₃ enabled efficient biomass hydrogenation. By utilizing detailed post-reaction characterization and in situ Raman studies, they track the formation of a sub-10 nm CdS layer under the operando regions where HMF hydrogenation takes place. This guarantees the formation of CdPS₃/CdS heterointerface, which accounts for the high FE observed in the reaction system. Moreover, density functional theory calculation further highlights the crucial role of in situ generated CdS layer in manipulating the adsorption energies of both HMF* and H*, as well as minimizing the intermediates energy barrier, thus accounting for the efficient HMF hydrogenation process. This work is solid and sound, offering fresh and inspiring insights that are valuable to the field, as it especially addresses the surface reconstruction phenomena in electrochemistry, which can be used for rational catalyst design not only for HMF hydrogenation but also for other reactions. Therefore, I recommend the acceptance of the manuscript in Nature Communications after considering the following points:

Response: We sincerely thank the reviewer for the favourable comments and approval of our manuscript. We have carefully addressed the comments made by the reviewer and revised the manuscript accordingly.

1. Minor points:

1) In the introduction the author states that “The rational design of an electrocatalyst for electrochemical conversion of biomass based feedstock requires a full understanding of the reaction process and the key parameters for controlling the electrode surface and interface.” please provide a reference for this statement.

Response: We thank the reviewer for the comment. The aforementioned sentence is cited by considering appropriate literature. Please see references 14 to 17 in the reference list for the details in our revised manuscript.

2) References are missed in the method section for the DFT calculation part. Some related refs should be cited here.

Response: All relevant citations has been made in the DFT calculation part of the method section. Please see references 50 to 56 in the reference list of our revised manuscript for the details.

3) XRD pattern in Fig. 2b, the broad peak assigned to the carbon cloth should be labeled.

Response: In Figure 2b, the broad peak at 2 theta of 27 degree belongs to carbon cloth substrate. We have now marked this peak with asterisk (*). The corresponding description has also been revised in the Figure caption. Please see Figure 2b of the revised manuscript, which is also reproduced in the response letter as follows:

Fig. 2b, XRD diffractogram of as prepared CdPS₃ on carbon cloth (cyan green) and the standard PDF card (PDF# 33-0243, red). The broad peak at ~ 27 degree (designated as *) belongs to the carbon cloth substrate.

2. In Figure 6, the authors tried to pair the in situ formed CdPS₃/CdS heterointerface cathode with MnCo₂O_{4.5}/Ni anode to synthesize two valuable chemicals. While detailed structural and electrochemical characterizations are offered for CdPS₃/CdS, some characterization for MnCo₂O_{4.5}/Ni catalyst (like XPS, EIS) should be provided.

Response: We sincerely appreciate this valuable suggestion. Following the reviewer suggestion, we have made an additional XPS and EIS characterization for the as synthesized MnCo₂O_{4.5}/NF electrocatalyst. Accordingly, we also revised the manuscript. For detailed information, please see the Supplementary Note II on lines 16 – 27, page S2 of the revised SI.

Supplementary Note II

“Structural characterizations of MnCo₂O_{4.5}/NF electrocatalyst. The MnCo₂O_{4.5}/NF were also characterized through the XPS analysis. The XPS survey spectrum of MnCo₂O_{4.5} sample clearly indicate the presence of Mn, Co, and O elements (Supplementary Fig. 32a). The high resolution XPS of Co 2p can be deconvoluted into a pair of peaks (Co 2p_{3/2} and 2p_{1/2}) that corresponds to the Co³⁺ (780.7 and 795.7 eV) and Co²⁺ (783.5 and 797.5 eV) along with two satellite peaks at 787.8 and 803 eV. This result suggests the co-existence of two oxidation states

of cobalt in the as-prepared $\text{MnCo}_2\text{O}_{4.5}$ (Supplementary Fig. 32b). Similarly, the high resolution XPS spectrum of Mn (Supplementary Fig. 32c) can be deconvoluted into a pair peak for Mn^{3+} (644.9 and 655.2 eV) and Mn^{2+} (642.2 and 653.6 eV), corresponding to Mn 2p_{3/2} and Mn 2p_{1/2}. The O 1s spectrum exhibit two distinct peaks at binding energies of 530 eV and 531.5 eV, which belongs to the M-O (Co/Mn-O) and hydroxyl species, respectively, on the surface of $\text{MnCo}_2\text{O}_{4.5}/\text{NF}$ (Supplementary Fig. 32d).”

Supplementary Fig. 32. a, Survey XPS spectra of the as-prepared $\text{MnCo}_2\text{O}_{4.5}/\text{NF}$ electrocatalyst. b-d, High resolution XPS spectra of Co 2p (b), Mn 2p (c) and O 1s (d).

Besides this, we also conducted EIS experiments by using the $\text{MnCo}_2\text{O}_{4.5}/\text{NF}$ electrode. The collected Nyquist plots in the presence and absence of 0.1M glycerol (Supplementary Fig. 34). The electrode exhibits lower charge transfer resistance (R_{ct}) of 5.7Ω when tested in the presence of 0.1M glycerol, which indicates faster electron transfer during the glycerol oxidation reaction than OER. These results suggest the enhanced electrocatalyst activity for GOR oxidation than that for OER. Accordingly, we have incorporated the following discussion in the revised manuscript. For detailed information, please see lines 20-23 on page 18 of the revised manuscript.

Lines 20-23 on Page 18.

“From the EIS measurement in Supplementary Fig. 34, the $\text{MnCo}_2\text{O}_{4.5}$ /NF electrode exhibit lower R_{ct} when tested in the presence of 0.1M glycerol (5.7Ω) than in only 1M KOH (26.1Ω), which indicates faster electron transfer during the GOR than OER.”

Supplementary Fig. 34. EIS of $\text{MnCo}_2\text{O}_{4.5}$ /NF in the presence of 0.1M glycerol (black) and in 1M KOH (red) at $1.6 V_{\text{RHE}}$.

3. When evaluating the performance of the electrode in Fig. 3, 10 mM HMF was utilized. Is this concentration the optimized one? If so, the electrocatalytic performances results related to other HMF concentration should be included.

Response: Thanks for the kind reminder. Actually, the concentration of 10 mM HMF was selected and used after carefully optimizing the catalytic activity of CdPS_3_{25} electrode under different initial concentration of HMF (such as 5, 10, 20, and 50 mM) to conduct the HMF hydrogenation reactions in Fig. 3. The HMF hydrogenation to BHMF performance of CdPS_3_{25} electrode in different HMF concentrations (Supplementary Fig. 34) shows the best FE for BHMF synthesis when 10 mM HMF is utilized as an initial concentration. It can be noted that an increase in HMF concentration to 20 mM and 50 mM leads to a decrease in FE towards BHMF synthesis, which could be probably due to the side reactions (formation of dimer or over hydrogenation) that consume reactants and intermediates (*ACS Catal.* 2020, 10, 19, 11643–11653, *Angew. Chem. Int. Ed.* 2022, 61). Besides, very low concentration of HMF (5 mM HMF) results in a relatively lower FE_{BHMF} which might be due to the competing HER and the catalyst behaviour. Based on the comment, we included this result as Supplementary Fig. 8 and the corresponding discussion is also incorporated in the main text. For detailed information, please see lines 15-17 on page 8 in the revised manuscript.

Lines 15-17 on page 8:

“Note that a maximum FE toward BHMF synthesis is achieved when the HMF concentration is 10 mM, while a further increase in HMF initial concentration to 50 mM significantly reduced FE_{BHMF} (Supplementary Fig. 8).”

Supplementary Fig. 8. The electrocatalytic HMF hydrogenation performance of CdPS₃_25 electrode using different initial concentration of HMF.

4. In Figure 3b, the authors found that the prior activation of the CdPS₃ electrode in the presence of HMF itself shows better performance for HMF hydrogenation to BHMF. This is interesting. I wonder if the same activation process can also result in some difference in the performance (FE) during the hydrogenation of other furan-based compounds (like furfural to furfuryl alcohol).

Response: We thank the reviewer for the insightful suggestion. To deeply explore whether the catalyst activation in the presence of other furan-based compounds have the same effect, we choose furfural as a model compound and examined the catalytic activity of CdPS₃ electrode for furfural hydrogenation. From the LSV curves in Supplementary Fig. 20a, we noticed that CdPS₃ electrode is active towards catalyzing furfural electro-reduction reaction. Next, we have evaluated the performance of this electrode for furfuryl alcohol synthesis *via* following the same activation processes utilized for HMF hydrogenation in Fig. 3b. As shown in Supplementary Fig. 20b-c, a clear difference in catalytic performance is observed when CdPS₃ electrode is activated under different CV cycles in the presence of furfural. The CdPS₃ electrode activated for 25 CV cycles in the presence of furfural still show better performance (FE ~ 63.6 % at -0.6 V_{RHE}) when compared with the others.

Supplementary Fig. 20. a, LSV curves of CdPS₃ electrode tested in 0.1 M PBS (blue) and in 0.1 M PBS + 20 mM furfural. b, Comparison of the electrocatalytic performance of CdPS₃ electrodes activated in 0.1 M PBS for 10 cycle, 0.1 M PBS, and 10mM HMF for 10 cycles, and 0.1 M PBS and 10mM HMF for 25 cycles @ towards furfuryl alcohol synthesis from furfural hydrogenation. The performance of these electrodes was evaluated at $-0.6 V_{RHE}$. c, the corresponding calibration curve for furfuryl alcohol quantification.

Following the aforementioned results, we further performed structural characterization on the CdPS₃_25 electrocatalyst after 3h furfural hydrogenation to furfuryl alcohol. The XRD pattern of this sample showed an extra peak corresponding to an *in situ* generated CdS species, which is similar to the phenomenon observed in the case of HMF hydrogenation. Besides, the Raman spectrum of this spent CdPS₃ electrode clearly depict a new Raman peak at 301cm^{-1} , which belongs to a CdS layer, which emanate from surface reconstruction. After carefully examining this experimental result, we reasoned that the furfural could also trigger the reconstruction of CdPS₃ nanosheets, which can also lead to an enhanced catalytic activity. Following the reviewer suggestion, we have revised the manuscript and included the following discussion and Figures. For detailed information, please see 21-30 on page 12 and lines 1-3 on Page 13 in the revised manuscript.

Lines 21-30 on page 12 and lines 1-3 on Page 13.

“To study whether the presence of other furan-based compounds could give rise to CdPS₃ surface reconstruction under similar activation process in Fig. 3b, we also examined the catalytic activity of furfural reduction on CdPS₃ electrode in 0.1M PBS. As shown in the LSV curves (Supplementary Fig. 20a), the CdPS₃ electrode is active towards furfural hydrogenation. Next, we examine the furfural hydrogenation catalytic performance on CdPS₃ electrodes activated for 10 CV cycles in only PBS, 10 CV cycles in PBS containing 10 mM HMF, and 25 cycles in PBS and 10 mM HMF. It is interesting to note that the catalyst activated via 25 CV cycles in 0.1 M PBS and 10 mM HMF display relatively higher FE of ~ 63.6 % at -

0.6 V_{RHE} (Supplementary Fig. 20b-c), which clearly shows the pivotal role of furfural in catalyst activation. Meanwhile, similar to that exhibited in HMF hydrogenation process, the post structural characterization of the spent electrode in furfural hydrogenation clearly depict the formation of CdPS₃/CdS heterostructure (Supplementary Fig. 21).”

Supplementary Fig. 21. Post structural characterizations of CdPS₃ electrode after furfural hydrogenation in PBS electrolyte at -0.6 V_{RHE} . a, Comparison of XRD pattern collected from the fresh and CdPS₃ electrode after reaction. b, The corresponding Raman peaks.

Reviewer 3

The manuscript presents a novel approach using layered CdPS₃ for biomass hydrogenation under considerably milder conditions than typical industrial processes. This study could potentially contribute significantly to the field due to its innovative method and the implications for sustainable energy production.

Strengths:

The concept of utilizing layered CdPS₃ for biomass hydrogenation at milder conditions is both innovative and relevant to current research trends in sustainable energy. The manuscript is well-structured, making it easier to understand the process and its potential implications.

Response: We would like to thank the reviewer for the positive comments and recommendation for publishing this work. We have addressed the comments and revised the manuscript accordingly.

Areas for Improvement:

Control Experiments: The central claim that the reconfiguration of CdPS₃ leads to the formation of CdS on the surface, which in turn acts as a hydrogenation platform, is intriguing. However, the manuscript currently lacks sufficient control experiments to robustly support this claim. To strengthen the findings:

Consider incorporating control experiments using CdS quantum dots, amorphous surfaces on CdPS₃, and other two-dimensional materials like graphene. These controls would help isolate the specific contributions of the CdPS₃ reconfiguration to the observed hydrogenation performance.

Response: We thank the reviewer for the insightful suggestion. We agree with the reviewer that verifying the role of the heterointerface in CdPS₃/CdS could provide further insight to our work. Following the suggestion, we synthesized a heterostructure *via* directly depositing CdS quantum dots (CdS_{QD}) on CdPS₃ nanosheets through one-step chemical bath deposition method to obtain CdPS₃/CdS_{QD} heterostructure (See Supplementary Fig. 26 in the revised manuscript). The XRD pattern of the as-synthesized sample clearly depicts the presence of cubic phase CdS (PDF#65-8873) without affecting the CdPS₃ nanosheet (Supplementary Fig. 26a). From the HR-TEM images in Supplementary Fig. 26b-c, a lattice fringe of ~ 0.2 and ~ 0.332 nm that corresponds to the (220) and (311) planes, respectively, of CdS is seen, which corroborate well with the XRD patterns in Supplementary Fig. 26a. These characterizations indicate that the CdS_{QD} are successfully decorated on the ultrathin CdPS₃ nanosheet surface. The electrocatalytic HMF hydrogenation performance of the as-fabricated electrocatalyst was then examined in 0.1M PBS electrolyte. Interestingly, the CdPS₃/CdS_{QD} electrocatalyst demonstrated a FE of 75.9%, which is better than the pristine CdS (maximum FE_{BHMF} = 44.8 % at -0.65 V_{RHE}) but lower than that of the *in situ* generated CdPS₃/CdS heterointerface electrode (91.3%). These results further highlight the crucial role of the *in situ* generated heterointerface for efficient BHMF synthesis. Following this experimental finding, we have incorporated the related discussion in the revised manuscript. Please see lines 14-23 on Page 13 and lines 25-32 on Page 20 for the details.

Lines 14-23 on Page 13:

“To further understand the role of the in situ generated CdS layer, we also synthesized a heterostructure through directly depositing CdS quantum dots on CdPS₃ nanosheets (CdPS₃/CdS_{QD}) using one-step chemical bath deposition method (Supplementary Fig. 26, See details in the experimental section). Interestingly, the as-fabricated CdPS₃/CdS_{QD} electrocatalyst demonstrate a FE_{BHMf} of 75.9% at 0.7 V_{RHE} (Supplementary Fig. 27), which is better than the pristine CdS (maximum FE_{BHMf} = 44.8 % at $-0.65 V_{RHE}$) but lower than that of the in situ generated CdPS₃/CdS heterointerface electrode (CdPS₃_25, FE_{BHMf} = 91.3%). These results further demonstrate the crucial role of the in situ generated heterointerface for efficient HMF hydrogenation.”

Lines 25-32 on Page 20.

“**Synthesis of CdPS₃/CdS_{QD} heterostructure catalyst.** To obtain the CdPS₃/CdS_{QD} heterostructure catalyst, the CdS quantum dots were directly deposited on the as-prepared CdPS₃ nanosheets by chemical bath deposition method. In a typical synthesis procedure, four beakers are first prepared; the first two contains 0.01M Cd(NO₃)₂ and 0.01M Na₂S (50 mL each), respectively, and the other two contains distilled water for rinsing the samples. Then, the CdPS₃/CC was sequentially immersed on the four beakers for 15 seconds. The process was repeated four times to obtain CdS quantum dots decorated CdPS₃ nanosheets (i.e., CdPS₃/CdS_{QD}).”

Supplementary Fig. 26. a-b, XRD patterns (a) and Raman spectrum (b) of the as-synthesized CdPS₃/CdS_{QD}. **c-d**, High resolution TEM images of the CdS quantum dots decorated on CdPS₃ nanosheets. In the XRD pattern, the peaks labelled with * belongs to the carbon cloth substrate.

Supplementary Fig. 27. a, LSV curves of CdPS₃/CdS_{QD} electrode tested in 0.1M PBS without and with 20 mM HMF. **b**, Potential dependent electrocatalytic HMF hydrogenation performance of as-synthesized CdPS₃/CdS_{QD} electrode.

Next, we rationally selected and synthesized other two representative 2D materials as control, that are In₂S₃ (Supplementary Fig. 22a,b) and CdPSe₃ nanosheets (Supplementary Fig. 23a).

First, from the LSV curves in Supplementary Fig. 22c, it can be seen that In₂S₃ nanosheet sample is active toward HMF reduction in 0.1 M PBS electrolyte. Next, we compared the HMF hydrogenation performance of In₂S₃ electrode after activating the In₂S₃ nanosheet electrode *via* 10 CV cycle in 0.1 M PBS, 10 CV cycle in 0.1 M PBS and 10 mM HMF, and 25 CV cycle in 0.1 M PBS and 10 mM HMF. Unlike the results observed in CdPS₃ nanosheet electrode, no apparent difference is seen in the FE_{BHMF} (varies only from ~45.5 to 46.5%) of the In₂S₃ nanosheet electrodes. The post structural and morphological characterizations shows that In₂S₃ nanosheet retained its structure and morphology after HMF hydrogenation test (Supplementary Fig. 23a-c) without any new peak being observed in its Raman spectra after HMF hydrogenation.

Second, we choose CdPSe₃ that possesses similar crystal structure with CdPS₃ as another control. The electrocatalytic activity of CdPSe₃ nanosheet electrode towards HMF hydrogenation was examined at different CV cycle activation. Different from the CdPS₃ nanosheet electrode, this electrocatalysts display almost similar FE values toward BHMF

synthesis irrespective to the CV activation in 0.1 M PBS ($FE_{\text{BHMF}} = 21.9\%$) and 0.1 M PBS and 10mM HMF electrolytes ($FE_{\text{BHMF}} = 20.3$ vs 21% for 10 and 25 CV activated samples). The post structural characterizations of CdPSe₃ nanosheet electrocatalyst after HMF hydrogenation catalysis did not present any evidence for the formation of CdSe species on the CdPSe₃ surface and no change is observed in CdPSe₃ nanosheets.

Overall, these control experiments clearly show that the difference in performance of CdPS₃ upon different activation process can be mainly attributed to the specific reconfiguration process, which is not seen in In₂S₃ and CdPSe₃ electrocatalysts. Following these experimental findings, we have incorporated the related discussion in the revised manuscript. Please see lines 3 -13 on page 13 for the details.

Lines 3 -13 on page 13.

“Next, we established a control experiment by employing other two-dimensional materials In₂S₃ and CdPSe₃ nanosheets (Supplementary Fig. 22-25). The HMF hydrogenation performance of these electrocatalysts was evaluated after activating them under different CV cycles. Unlike the CdPS₃ electrode, both In₂S₃ and CdPSe₃ nanosheets did not show appreciable change in the catalytic performance under different CV activation process. From the post-reaction characterizations, there is no change in the catalyst structure and surface after prolonged operation, which evidence the absence of surface reconstruction during HMF hydrogenation process on In₂S₃ and CdPSe₃ nanosheet electrode. The control experiments clearly show that the specific reconfiguration of CdPS₃ is the main attributing factor to the promoted performance for hydrogenation of HMF.”

Supplementary Fig. 22. Morphology characterizations and electrocatalytic activity of In₂S₃ nanosheet catalyst for hydrogenation of HMF to BHMF. a-b, SEM images of In₂S₃ nanosheets grown on carbon cloth substrate. c, LSV curves of In₂S₃ nanosheet electrode in 0.1 M PBS (pH= 9.2) electrolyte with and without 20 mM HMF 0.1 M PBS. d, Comparison of HMF hydrogenation activity of In₂S₃ nanosheet electrodes activated in 0.1 M PBS in the presence and absence of HMF under 10 and 25 CV activation cycles.

Supplementary Fig. 23. a, comparison of In₂S₃ nanosheets before and after HMF hydrogenation tests using Raman characterizations. b-c, The corresponding SEM images of In₂S₃_25 nanosheets after electrocatalytic HMF hydrogenation test.

Supplementary Fig. 24. Morphology characterizations and electrocatalytic activity of CdPSe₃ nanosheet catalyst for hydrogenation of HMF to BHMF. a, SEM images of CdPSe₃ nanosheets grown on carbon cloth substrate. b, LSV curves of CdPSe₃ nanosheet electrode directly synthesized on carbon cloth substrate using SCCVC method in 0.1 M PBS (pH= 9.2) electrolyte with (black) and without (red) 20 mM HMF. c, Comparison of HMF hydrogenation activity of CdPSe₃ nanosheet electrodes activated in 0.1 M PBS in the presence and absence of HMF under 10 and 25 CV activation cycles.

Supplementary Fig. 25. Characterization of CdPSe₃_25 nanosheet before and after HMF hydrogenation reaction test. a) XRD patterns. b) Raman spectra.

Clarification of Claims: While the degradation of CdPS₃ to form CdS is a central aspect of the study, it would be beneficial to elaborate on how this degradation process directly contributes to the enhanced hydrogenation performance. Additional experimental data supporting this mechanism would add depth to the study.

Response: We thank the reviewer for the comment. Detailed study that systematically examine the mechanism how a specific catalyst works would offer an inspiring perspective to the readers. Earlier works on surface reconstruction highlighted that the adsorption /desorption of

reactants and vital intermediates could be greatly influenced by the as-formed composited structure of a reconstructed surface (Chem. Soc. Rev., 2021, 50, 8428–8469). The theoretical calculations showed that the Cd of the in situ generated CdS and S site at the interface of CdPS₃/CdS are the active sites for adsorption of HMF* and H*, respectively. The *in-situ* generated interface could simultaneously enhance the HMF adsorption on Cd sites and H adsorption on S sites nearby to complete the hydrogenation process. From this point, it can be understood that a relatively larger or denser reconstructed CdS layer rather than thicker CdS layer on CdPS₃ nanosheet would avail more Cd and S (at interface) active sites that favour the hydrogenation process.

Besides, based on the comment, we have highlighted the catalyst activation process in the presence of the reactant itself (HMF) or furan-group to facilitate the specific reconfiguration on CdPS₃ surface. This reconstructed CdPS₃ electrode not only catalyze the HMF hydrogenation to BHMF, but also facilitate furfural electro-reduction to furfuryl alcohol (Supplementary Fig. 20a). Next, we have evaluated the performance of this electrode for furfuryl alcohol synthesis via following the same activation processes utilized for HMF hydrogenation in Fig. 3b. As shown in Supplementary Fig. 20b-c, a clear difference in catalytic performance is observed when CdPS₃ electrode is activated under different CV cycles in the presence of furfural substrate. The CdPS₃ electrode activated for 25 CV cycles in the presence of furfural still show better performance when compared with the others.

Supplementary Fig. 20. a, LSV curves of CdPS₃ electrode tested in 0.1 M PBS (blue) and in 0.1 M PBS + 20 mM furfural. b, Comparison of the electrocatalytic performance of CdPS₃ electrodes activated in 0.1 M PBS for 10 cycle, 0.1 M PBS, and 10mM HMF for 10 cycles, and 0.1 M PBS and 10mM HMF for 25 cycles towards furfuryl alcohol synthesis from furfural hydrogenation. The performance of these electrodes was evaluated at -0.6 V_{RHE}. c, the corresponding calibration curve for furfuryl alcohol quantification.

Following the aforementioned results, we further performed structural characterization on the CdPS₃_25 electrocatalyst after 3h furfural hydrogenation to furfuryl alcohol. The XRD pattern of this sample showed an extra peak corresponding to an *in situ* generated CdS species, which is similar to the phenomenon observed in the case of HMF hydrogenation. Besides, the Raman spectrum of this spent CdPS₃ electrode clearly depict a new Raman peak at 301cm⁻¹, which belongs to a CdS layer, which emanate from surface reconstruction. After carefully examining this experimental result, we reasoned that the furfural could also trigger the reconstruction of CdPS₃ nanosheets, which can also lead to an enhanced catalytic activity. Following the reviewer suggestion, we have revised the manuscript and included the following discussion and Figures. For detailed information, please see 21-30 on page 12 and lines 1-3 on Page 13 in the revised manuscript.

Lines 21-30 on page 12 and lines 1-3 on Page 13.

“To study whether the presence of other furan-based compounds could give rise to CdPS₃ surface reconstruction under similar activation process in Fig. 3b, we also examined the catalytic activity of furfural reduction on CdPS₃ electrode in 0.1M PBS. As shown in the LSV curves (Supplementary Fig. 20a), the CdPS₃ electrode is active towards furfural hydrogenation. Next, we examine the furfural hydrogenation catalytic performance on CdPS₃ electrodes activated for 10 CV cycles in only PBS, 10 CV cycles in PBS containing 10 mM HMF, and 25 cycles in PBS and 10 mM HMF. It is interesting to note that the catalyst activated via 25 CV cycles in 0.1 M PBS and 10 mM HMF display relatively higher FE of ~ 63.6 % at -0.6 V_{RHE} (Supplementary Fig. 20b-c), which clearly shows the pivotal role of furfural in catalyst activation. Meanwhile, similar to that exhibited in HMF hydrogenation process, the post structural characterization of the spent electrode in furfural hydrogenation clearly depict the formation of CdPS₃/CdS heterostructure (Supplementary Fig. 21).”

Supplementary Fig. 21. Post structural characterizations of CdPS₃ electrode after furfural hydrogenation in PBS electrolyte at -0.6 V_{RHE}. a) Comparison of XRD pattern collected from the fresh and CdPS₃ electrode after reaction. b) the corresponding Raman peaks.

Lines 4-7 on Page 18.

“...adsorption energetics of H and HMF*. The in-situ generated interface could simultaneously enhance the HMF adsorption on Cd sites and H adsorption on S sites nearby to complete the hydrogenation process with a lower energy barrier.”*

Conclusion:

The manuscript presents a promising approach to biomass hydrogenation using layered CdPS₃. However, to fully substantiate the claims and provide a comprehensive understanding of the process, additional control experiments are essential. Incorporating these experiments and providing more detailed explanations of the underlying mechanisms would greatly enhance the manuscript's contribution to the field. In its current state, I would recommend a reject-resubmit before considering the manuscript for publication.

Response: We thank the reviewer for evaluating our manuscript and giving us valuable suggestions. We believe that the additional control experiments and discussion in the revised manuscript have significantly improved the quality of our work.

REVIEWERS' COMMENTS

Reviewer #1 (Remarks to the Author):

The authors have appropriately addressed my comments. I have also reviewed their response with respect to the other referees' comments. I am impressed by the new control experiments and the related discussions they supplied to address these issues. I think the manuscript now has an significantly improved quality and thus recommend it for publication as is.

Reviewer #2 (Remarks to the Author):

My comments have been addressed properly. I recommend accept.

Point-by-point reply to the reviewers' comments

Reviewer 1

The authors have appropriately addressed my comments. I have also reviewed their response with respect to the other referees' comments. I am impressed by the new control experiments and the related discussions they supplied to address these issues. I think the manuscript now has an significantly improved quality and thus recommend it for publication as is.

Response: We sincerely thank the reviewer for his/her positive comments and approval our manuscript for publication in *Nature communications*.

Reviewer 2

My comments have been addressed properly. I recommend accept.

Response: We sincerely thank the reviewer for the positive assessment and approval of our manuscript for publication.